# SARS-CoV-2 ORF6 disrupts innate immune signalling by inhibiting cellular mRNA export

**Ross Hall**[1ᵒ], **Anabel Guedán**[1ᵒ], **Melvyn W. Yap**[1ᵒ], **George R. Young**[2], **Ruth Harvey**[3], **Jonathan P. Stoye**[4,5], **Kate N. Bishop**[1]*

1 Retroviral Replication Laboratory, The Francis Crick Institute, London, United Kingdom, 2 Bioinformatics and Biostatistics STP, The Francis Crick Institute, London, United Kingdom, 3 World Influenza Centre, The Francis Crick Institute, London, United Kingdom, 4 Retrovirus-Host Interactions Laboratory, The Francis Crick Institute, London, United Kingdom, 5 Department of Infectious Disease, Imperial College London, United Kingdom

ᵒ These authors contributed equally to this work.
* kate.bishop@crick.ac.uk

**Data Availability Statement:** All relevant data are within the manuscript and its Supporting Information files. The raw RNAseq data from this

## Abstract

SARS-CoV-2 is a betacoronavirus and the etiological agent of COVID-19, a devastating infectious disease. Due to its far-reaching effect on human health, there is an urgent and growing need to understand the viral molecular biology of SARS-CoV-2 and its interaction with the host cell. SARS-CoV-2 encodes 9 predicted accessory proteins, which are presumed to be dispensable for *in vitro* replication, most likely having a role in modulating the host cell environment to aid viral replication. Here we show that the ORF6 accessory protein interacts with cellular Rae1 to inhibit cellular protein production by blocking mRNA export. We utilised cell fractionation coupled with mRNAseq to explore which cellular mRNA species are affected by ORF6 expression and show that ORF6 can inhibit the export of many mRNA including those encoding antiviral factors such as IRF1 and RIG-I. We also show that export of these mRNA is blocked in the context of SARS-CoV-2 infection. Together, our studies identify a novel mechanism by which SARS-CoV-2 can manipulate the host cell environment to supress antiviral responses, providing further understanding to the replication strategies of a virus that has caused an unprecedented global health crisis.

## Author summary

SARS-CoV-2 is the virus responsible for the current COVID-19 pandemic. Coronaviruses, like SARS-CoV-2, replicate their genome in the cytoplasm of the host cell by hijacking the cellular machinery. In addition to structural proteins and viral enzymes, SARS-CoV-2 is predicted to encode 9 accessory proteins. Although these are not required for *in vitro* replication, they are thought to modulate the host cell environment to favour viral replication. In this work, we show that the ORF6 accessory protein can supress cellular protein production by blocking mRNA nuclear export through interacting with the cellular protein Rae1, a known mRNA export factor. We also investigated which cellular mRNAs were retained in the nucleus when ORF6 was overexpressed. Interestingly, we found that ORF6 inhibited the export of many different mRNAs, including those

study can be accessed from ENA via accession PRJEB49943.

**Funding:** This work was supported by the Francis Crick Institute (https://www.crick.ac.uk), which receives its core funding from Cancer Research UK (FC001042, to KB and FC001162, to JS), the UK Medical Research Council (FC001042, to KB and FC001162, to JS) and the Wellcome trust (FC001042, to KB and FC001162, to JS) and by the MRC Genotype-to-Phenotype (G2P) UK National Virology Consortium grant to KB (MR/W005611/1) and by a Wellcome Trust Investigator Award to JS (108012/Z/15/Z). The funders had no role in study design, data collection and analysis, decision to publish, or preparation of the manuscript.

**Competing interests:** The authors have declared that no competing interests exist.

encoding antiviral factors, like IRF1 and RIG-I, even in the absence of stimulation by interferon. Importantly, we found that the export of these mRNAs was similarly affected in the context of SARS-CoV-2 infection. Therefore, we believe we have identified a novel mechanism that SARS-CoV-2 uses to suppress antiviral responses in order to make the cell more permissive to infection.

## Introduction

Severe acute respiratory syndrome coronavirus 2 (SARS-CoV-2) is an enveloped single-stranded, positive-sense RNA virus of the *Betacoronovirus* genus and is the etiological agent of the highly infectious disease, COVID-19. SARS-CoV-2 replicates in the epithelial cells of the respiratory tract causing extensive respiratory symptoms, with severe cases leading to mortality [1–4]. SARS-CoV-2 expresses four structural proteins, spike (S), membrane (M), envelope (E) and nucleoprotein (N) and 16 non-structural proteins (nsps). It also encodes nine predicted accessory proteins; ORF3a, ORF3d, ORF6, ORF7a, ORF7b, ORF8, ORF9b, ORF9c and ORF10 [2,5–7], although there is little experimental evidence of protein-coding function for ORF9c or ORF10 [8]. These accessory proteins are presumed dispensable for *in vitro* replication and have been suggested to play a role in pathogenicity and host cell modulation [9,10]. Little is known about the function of these accessory proteins and how they affect the cellular environment to facilitate efficient SARS-CoV-2 replication.

SARS-CoV-2 shares a high degree of homology with the etiological agent of the 2003 SARS outbreak, SARS-CoV (hereafter known as SARS-CoV-1). Indeed, the accessory proteins of these two betacoronaviruses have high homology. ORF6 from SARS-CoV-1 is a 7.3 kDa protein, which was previously reported to inhibit IFN signalling by disrupting the nuclear import of STAT1/STAT2 and IRF3 [11,12]. SARS-CoV-2 ORF6 has also been reported to inhibit IFN signalling and this is dependent on ORF6 interacting with the cellular Rae1-Nup98 complex [13,14], which spans the nuclear envelope, providing a gateway between the nucleus and cytoplasm, and helps shuttle proteins and mRNA between the two compartments [15,16]. The Rae1-Nup98 complex also has a role in the cell cycle [17–19]. Other viruses disrupt this nuclear export pathway to aid viral propagation. VSV matrix (M) and herpesvirus ORF10 both inhibit mRNA nuclear export by disrupting Rae1-Nup98 function, while Influenza virus NS1 renders cells more permissive to replication by downregulating Nup98 expression thereby inhibiting Rae1-dependent mRNA nuclear export [20–23]. As coronaviruses transcribe their genome in the cytoplasm, they are not dependent on nuclear export function for viral replication. Indeed, NSP1 proteins from coronaviruses have been shown to inhibit the NXF1 mRNA export pathway, potentially disrupting the expression of a range of cellular proteins [24].

Given the potential importance of the SARS-CoV-2 accessory proteins we set out to investigate their functions. Strikingly, we found that ORF6 inhibited protein production and we show that in addition to disrupting STAT1/2 import, ORF6 can also block the export of cellular mRNA from the nucleus into the cytoplasm, effectively blocking protein translation. In agreement with other very recent studies of SARS-CoV-2 ORF6, this is also dependent on interaction with the Rae1-Nup98 complex [13,14,25]. Here, we utilise cellular fractionation and mRNAseq to identify the mRNA species blocked by ORF6. We demonstrate that ORF6 inhibits export of a broad range of mRNAs, in particular several interferon-upregulated genes that encode antiviral factors, including RIG-I and IRF1. This shows that SARS-CoV-2 ORF6 can disrupt host cell innate immune signalling by blocking mRNA nuclear export and suggests that ORF6 may function to inhibit the earliest stages of innate signalling by downregulating

expression of pathogen recognition receptors (PRRs) and antiviral transcription factors, such as IRF1.

## Results

### SARS-CoV-2 ORF6 inhibits protein expression

To investigate the role of SARS-CoV-2 accessory proteins, we first generated a panel of HIV-1 virus-like-particles (VLPs) encoding each open-reading frame (ORF) to transduce a range of different cell lines to express each accessory protein individually. VLPs were synthesised by co-transfecting HEK293T cells with plasmids encoding HIV-1 Gag-Pol (pCMVΔ8.91), VSV-G (pVSV-G) and a pLVX-StrepII-ORF vector, encoding either ORF3a, ORF3d, ORF6, ORF7a, ORF7b, ORF8, ORF9b, ORF9c or ORF10. VLPs titres were determined after 72h and at least $10^4$ infectious units/ml were detected for all VLPs except for VLP encoding ORF6 which had undetectable infectious units (Fig 1A, upper panel). To determine if this was due to decreased infectivity or decreased particle production, transfected HEK293T cell lysates were analysed by immunoblotting for the HIV-1 structural polyprotein, Gag. Gag was detectable in all cells, except for cells transfected with pLVX-StrepII-ORF6 (Fig 1A, lower panel). This suggested that ORF6 expression resulted in inhibition of Gag production preventing formation of VLPs.

To determine if there was a dose-dependent effect on Gag expression and VLP production, we next generated GFP-reporter HIV-1 VLPs in the presence of increasing amounts of ORF6 or ORF9b as a control. Transfected HEK293T cell lysates were analysed by immunoblotting for HIV-1 Gag and ORF6/ORF9b. Gag expression was reduced in a dose-dependent manner only with increasing amounts of ORF6 (Fig 1B). VLPs generated from these transfections were used to transduce HeLa cells and the percentage of cells expressing GFP was determined after 72h by flow cytometry to measure VLP titre. As with Gag expression, HIV-1 VLP titres were severely reduced in a dose dependent manner compared to the control when produced in the presence of ORF6 (Fig 1C, yellow lines) but not ORF9b (Fig 1C, blue lines).

To determine if ORF6 inhibition was specific to HIV-1 Gag expression, we next examined the effect of ORF6 on the expression of MLV Gag and mApple fluorescent protein, which have completely different protein sequences to HIV-1 Gag. As with HIV-1 Gag, ORF6 inhibited MLV Gag expression and VLP production in a dose-dependent manner (S1A and S1B Fig). In order to detect all the SARS-CoV-2 ORFs using the same antibody (anti-strep), and to increase ORF expression levels, we decided to switch expression vectors to using a pLVX-EF1α-SARS-CoV-2- ORF-2xStrep/2xStrep-ORF -IRES-Puro vector (referred to as twin-strep tag), kindly provided by Nevan Krogan. We confirmed that ORF6 expressed from this vector inhibited Gag expression and viral infectivity as strongly as we had seen before (S1C Fig). Furthermore, when HEK293T cells were co-transfected with plasmids expressing mApple and each SARS-CoV-2 ORF in the twin-strep tagged vector, and the median fluorescent intensity (MFI) of mApple measured by flow cytometry, ORF6 reduced the MFI approximately 3-fold compared to the control (Fig 1D, upper panel). None of the other ORFs had an effect on the MFI of mApple (Fig 1D, upper panel), although immunoblotting revealed very low expression of ORF3d, ORF9c and ORF10 (Fig 1D, lower panel), which has also been observed by others [5]. Taken together our results suggest that ORF6 can inhibit expression of multiple proteins expressed from a vector.

Previous work has shown that the C-terminal region of ORF6 is important for inhibiting IRF3 and STAT nuclear import [5,11,14]. To determine if this region is also important for ORF6-dependent inhibition of protein expression, we introduced alanine substitutions at residues in the C-terminus that were conserved between SARS-CoV-2 and SARS-CoV-1 ORF6 proteins, E55A, M58A, E59A or D61A (Fig 1E, highlighted in red). HEK293T cells were co-

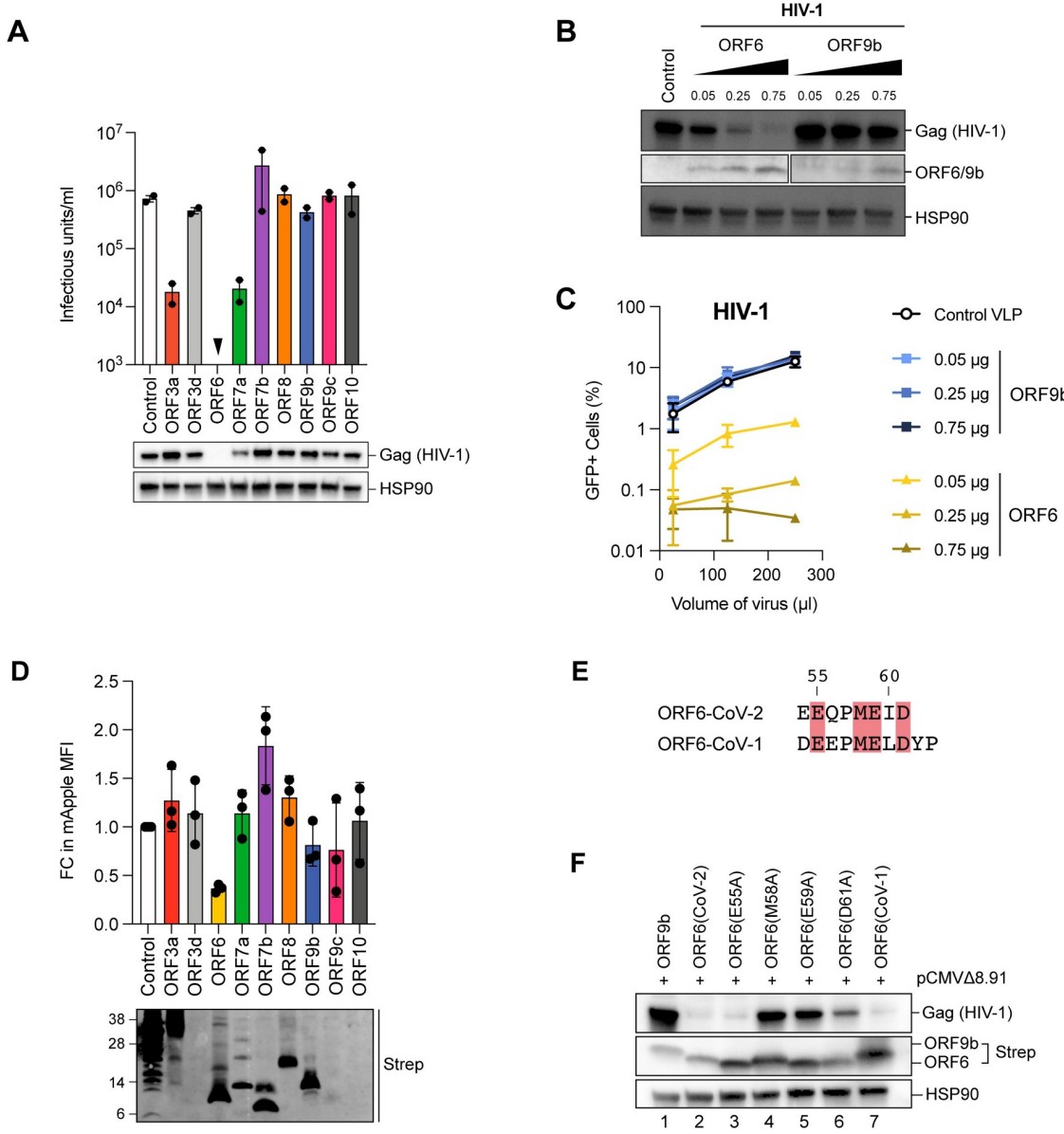

**Fig 1. SARS-CoV-2 ORF6 inhibits protein expression.** (A) To generate VLPs, HEK293T cells were co-transfected with plasmids encoding HIV-1 Gag-Pol, VSV-G and a pLVX-StrepII-ORF vector encoding either SARS-CoV-2 ORF3a, ORF3d, ORF6, ORF7a, ORF7b, ORF8, ORF9b, ORF9c or ORF10 or an empty vector control. VLPs were harvested and titrated in HeLa cells. The infectious units/ml are shown in the bar graph. Points indicated independent biological repeats. Transfected cell lysates were separated by SDS-PAGE and analysed for Gag(HIV-1) and HSP90 by immunoblotting. (B,C) HEK293T cells were transfected with plasmids encoding HIV-1 Gag-Pol, VSV-G and either increasing amounts of pLVX-StrepII-ORF6 or pLVX-StrepII-ORF9b as well as a GFP reporter plasmid, pCSGW. (B) Transfected cell lysates were separated by SDS-PAGE and analysed for Gag(HIV-1), ORF6 or ORF9b and HSP90 by immunoblotting. (C) HeLa cells were infected with increasing amounts of HIV-1 VLPs. Three days post infection, the percentage of GFP positive cells was determined by flow cytometry. Graph shows the mean and range of two biological repeats. (D) HEK293T cells were transfected with the pLVX-EF1α plasmid encoding strep-tagged GFP(Control), ORF3a, ORF3d, ORF6, ORF7a, ORF7b, ORF8, ORF9b, ORF9c or ORF10 and a plasmid encoding mApple. After 48h, cells were fixed and the mApple median fluorescent intensity (MFI) measured by flow cytometry. The fold change (FC) in MFI is plotted relative to the control for independent biological repeats. Error bars show SEM. Transfected cell lysates were analysed by SDS-PAGE and immunoblotting with anti-strep. (E) Amino acid alignment of the C-terminus region of ORF6 from SARS-CoV-2 and SARS-CoV-1 with conserved residues highlighted in red. (F) HEK293T cells were transfected with pLVX-EF1α encoding strep-tagged ORF9b, ORF6(CoV-2), ORF6(CoV-1) or indicated ORF6(CoV-2) mutant and a plasmid encoding HIV-1 Gag-Pol. After 24h, cell lysates were separated by SDS-PAGE and analysed for Gag(HIV-1), Strep and HSP90 by immunoblotting.

transfected with plasmids encoding each mutant ORF6 or wild type ORF6 from either SARS-CoV-2 or SARS-CoV-1 together with HIV-1 Gag and cell lysates were analysed by immunoblotting (Fig 1F). As before, ORF6(CoV-2) (Fig 1F, lane 2) reduced Gag expression compared to the ORF9b control (lane 1). In addition, ORF6(CoV-1) (lane 7) and mutant E55A (lane 3) were also able to inhibit Gag expression. However, ORF6 mutants M58A (lane 4), E59A (lane

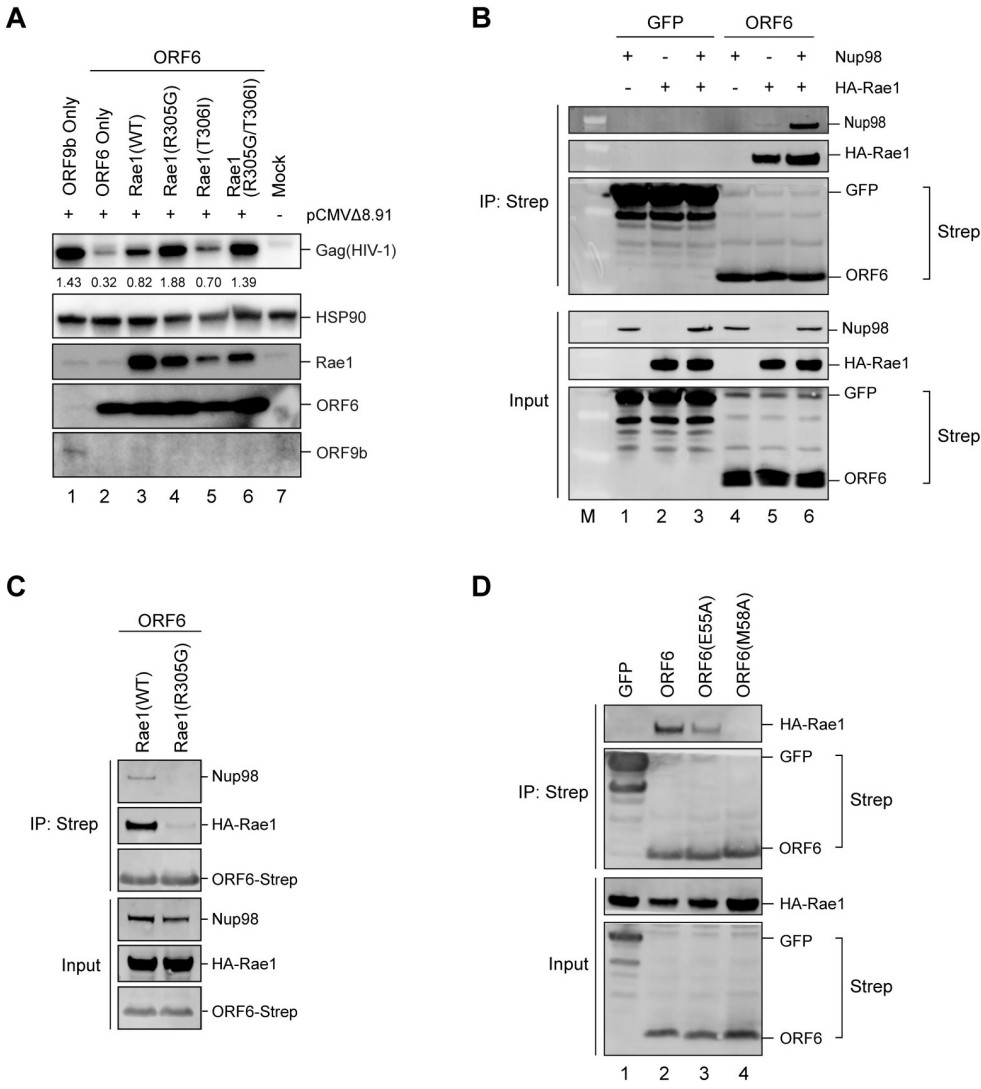

**Fig 2. ORF6 inhibits protein expression by interfering with Rae1.** (A) HEK293T cells were co-transfected with plasmids encoding HIV-1 Gag-Pol, ORF6 or ORF9b and Rae1 or indicated Rae1 mutant. Mock cells were untransfected. After 48h, cell lysates were separated by SDS-PAGE and analysed for Gag(HIV-1), Rae1, ORF6 or ORF9b and HSP90 by immunoblotting. Gag expression was quantified, normalised to HSP90, and shown in numbers below the Gag panel. (B) HEK293T cells were transfected with plasmids encoding Twin-Strep-tagged GFP or ORF6 and Nup98 and/or HA-Rae1. After 24h, Twin-Strep-tagged proteins were immunoprecipitated with MagStrep beads and proteins eluted with biotin. Input lysates and eluate were separated by SDS-PAGE and analysed by immunoblotting for Strep, Nup98 and HA. (C) HEK293T cells were transfected with plasmids encoding Twin-Strep-tagged GFP or ORF6, Nup98 and either HA-Rae1(WT) or HA-Rae1(R305G). After 24h, Twin-Strep-tagged proteins were immunoprecipitated and analysed as in (C). (D) HEK293T cells were transfected with plasmids encoding Twin-Strep-tagged proteins; ORF6(CoV-2), ORF6(E55A), ORF6(M58A) or GFP and HA-Rae1(WT). After 24h, Twin-Strep-tagged proteins were immunoprecipitated with MagStep beads and input lysates and eluates separated by SDS-PAGE and analysed by immunoblotting for Strep and HA.

5) and D61A (lane 6) had a much weaker effect on Gag expression suggesting that residues M58, E59 and D61 are essential for reducing protein expression. ORF6 mutants M58A, E59A and D61A also had a much weaker effect on mApple expression compared to wild type ORF6 (S1D Fig).

## ORF6 inhibits protein expression by interfering with Rae1

There are several ways that ORF6 could block protein expression, for example inhibiting transcription, mRNA nuclear export or translation. Biochemical studies have indicated that SARS-CoV-2 ORF6 interacts with the Rae1-Nup98 complex [5] which is important for mRNA nuclear export. Specifically, Rae1 helps shuttle nascent mRNA into the cytoplasm for translation [15,16,26]. Therefore, to investigate whether ORF6 could block protein expression by interfering with the Rae1-Nup98 complex, we overexpressed Rae1 in HEK293T cells together with HIV-1 Gag and ORF6. Here, we used the original ORF6 expression vector to keep the protein levels of ORF6 in line with the Rael levels. Fig 2A shows that expressing exogenous Rae1 partially rescued Gag expression in the presence of ORF6 (Fig 2A, compare lanes 2 and 3). Relief from ORF6 inhibition could be further enhanced by introduction of a mutation known to affect the inhibitory interaction between the VSV M protein and Rae1 [20]. Whilst expression of Rae1(T306I) had a similar effect to overexpressing wild type Rae1 (lane 5), expression of Rae1(R305G) fully rescued Gag expression to the levels seen in the ORF9b control (compare lane 4 with lane 1). This suggests that ORF6 interacts with Rae1 to inhibit protein expression and that R305 is important for the interaction.

To confirm the interaction of ORF6 with Rae1 and examine the interaction with Nup98, we performed co-immunoprecipitations. HEK293T cells were co-transfected with plasmids expressing Strep-tagged GFP or ORF6 with Nup98 and/or HA-Rae1. After 24h, cells were lysed and GFP or ORF6 immunoprecipitated with anti-Strep antibody. Input lysates and precipitated proteins were analysed by immunoblotting for Strep, HA and Nup98. Both GFP and ORF6 were detected in the eluate, confirming successful immunoprecipitation (Fig 2B). The GFP control did not co-immunoprecipitate with either HA-Rae1 or Nup98. However, HA-Rae1 was detectable in the eluate when co-expressed with ORF6, confirming an interaction between these two proteins. Rae1 was detectable whether Nup98 was co-expressed or not, suggesting that the ORF6 interaction with Rae1 is not dependent on Nup98. Conversely, Nup98 was only detectable in the eluate when co-expressed with both ORF6 and Rae1. ORF6 and Rae1 also co-immunoprecipitated with a small amount of endogenous Nup98 (Fig 2B, lane 5). This suggests that the interaction between Nup98 and ORF6 is not direct but is instead dependent on Rae1. Importantly, the Rae1(R305G) mutation that conferred resistance to ORF6-dependent inhibition of protein expression (Fig 2A) did not co-immunoprecipitate with ORF6 (Fig 2C) and Nup98 was also not present under these conditions, confirming that Nup98 interacts with ORF6 via Rae1. We also tested whether the ORF6 mutants could co-precipitate Rae1 (Fig 2D). As expected, ORF6 and ORF6(E55A), which both inhibited Gag expression (Fig 1F), co-immunoprecipitated with Rae1 whilst ORF6(M58A) did not (Fig 2D). Additionally, we also observed that ORF6(CoV-1) co-immunoprecipitated with Rae1 (S2A Fig), and that both ORF6 homologues co-immunoprecipitated with endogenous Rae1 (S2B Fig). Overall, this implies that the effect of ORF6 on protein expression is mediated through an interaction with Rae1, suggesting that the block is to mRNA export.

## ORF6 inhibits the nuclear export of cellular mRNA

To investigate the effect of ORF6 on mRNA export directly, we first set up an assay to separate nuclear (Nucl) and cytoplasmic (Cyto) fractions and measure mRNA levels in each by qPCR.

A higher ratio of Nucl:Cyto mRNA compared to the control would suggest that the mRNA is being retained in the nucleus. As a proof-of-principle, we treated cells with Leptomycin B (LMB), which inhibits Chromosomal Maintenance 1 (CRM1)-dependent mRNA export [27] (S3 Fig). HEK293T cells were transfected with a plasmid expressing GFP, treated with LMB for 16h and then fractionated. LMB clearly reduced GFP expression in cells (S3A and S3B Fig). Absolute levels of GFP mRNA were determined by qPCR in the Nucl and Cyto fractions (S3C Fig). Although total GFP RNA levels were reduced, LMB-treated cells also had a higher Nucl: Cyto GFP mRNA ratio compared to untreated cells, at 0.93 compared to 0.35 (S3D Fig). Like LMB-treated cells, cells expressing ORF6 also reduced GFP expression (S3E and S3F Fig) and increased the ratio of Nucl:Cyto GFP mRNA (S3G and S3H Fig), while cells expressing ORF6 (M58A) had a similar Nucl:Cyto GFP mRNA as the control cells expressing ORF9b, reflecting the only minor reduction in GFP expression (S3E and S3F Fig). Overall, this indicates that ORF6 inhibits nuclear export of GFP mRNA, in a Rae1-dependent manner.

So far, we have shown that ORF6 can inhibit exogenous protein mRNA nuclear export and that this likely accounts for the block to protein production. To investigate whether ORF6 also inhibits endogenous cellular mRNA export, and to determine whether specific proteins are affected, we performed mRNAseq on nuclear and cytoplasmic fractions of HEK293T cells either untransfected (mock) or transfected with ORF6 or GFP. Cells were also treated with IFN-β to stimulate expression of interferon stimulated genes (ISGs) to determine if ORF6 can affect the export of these genes. Transfections were performed in duplicate and after 24h, one sample for each transfection was treated with IFN-β (1,000 units/ml) for 16h. Cells were then fractionated, and a sample was taken for protein analysis. RNA was extracted from the remaining fraction and mRNA libraries were prepared and sequenced (Fig 3A). Lysates were analysed by immunoblotting with anti-Strep antibodies to confirm ORF6/GFP expression, HSP90 (cytoplasmic marker) and lamin B1 (nuclear marker), to confirm that cellular fractionation was successful and phosphorylated STAT1 (pSTAT-Y701) to confirm activation of the type-I IFN signalling cascade (S4A Fig).

Next, we determined the distribution of mRNA between the cytoplasm and nucleus during ORF6 and GFP expression without and with IFN-β treatment (S4B Fig, S1 Table). ORF6 was then further compared to GFP to determine which mRNA species were specifically enriched in the nucleus or cytoplasm in the presence of ORF6 (Fig 3B, S1 Table). A positive fold change (FC) indicates that an mRNA species is enriched in the nucleus, while a negative FC suggests that an mRNA species is enriched in the cytoplasm. In total, in the absence of IFN, 2,431 mRNA species were enriched in the nucleus with ORF6, while only 180 mRNA species were enriched in the cytoplasm, compared to the control (Fig 3B, left panel, highlighted in yellow). With IFN-β treatment, 2,128 mRNA species were enriched in the nucleus with ORF6, while 81 were enriched in the cytoplasm, compared to the control (Fig 3B, right panel). There was considerable overlap (1,765 mRNA) between the mRNA species enriched in the nucleus in both conditions (Fig 3C). These genes were found to include a wide range of Gene Ontology (GO) terms (Fig 3D), highlighting their breadth of function. These data suggest that ORF6 not only inhibits mRNA export of exogenous mRNA such as GFP but, more importantly, also inhibits the nuclear export of a broad range of cellular mRNA.

## ORF6 inhibits the nuclear export of IFN-upregulated mRNA

As many mRNA were retained in the nucleus, we decided to investigate whether genes with potential anti-viral functions were among them. Firstly, we identified potential ISGs in our dataset by determining which mRNA species were upregulated by IFN-β treatment in cells transfected with our control plasmid (S5A Fig). We found 134 mRNAs species that were

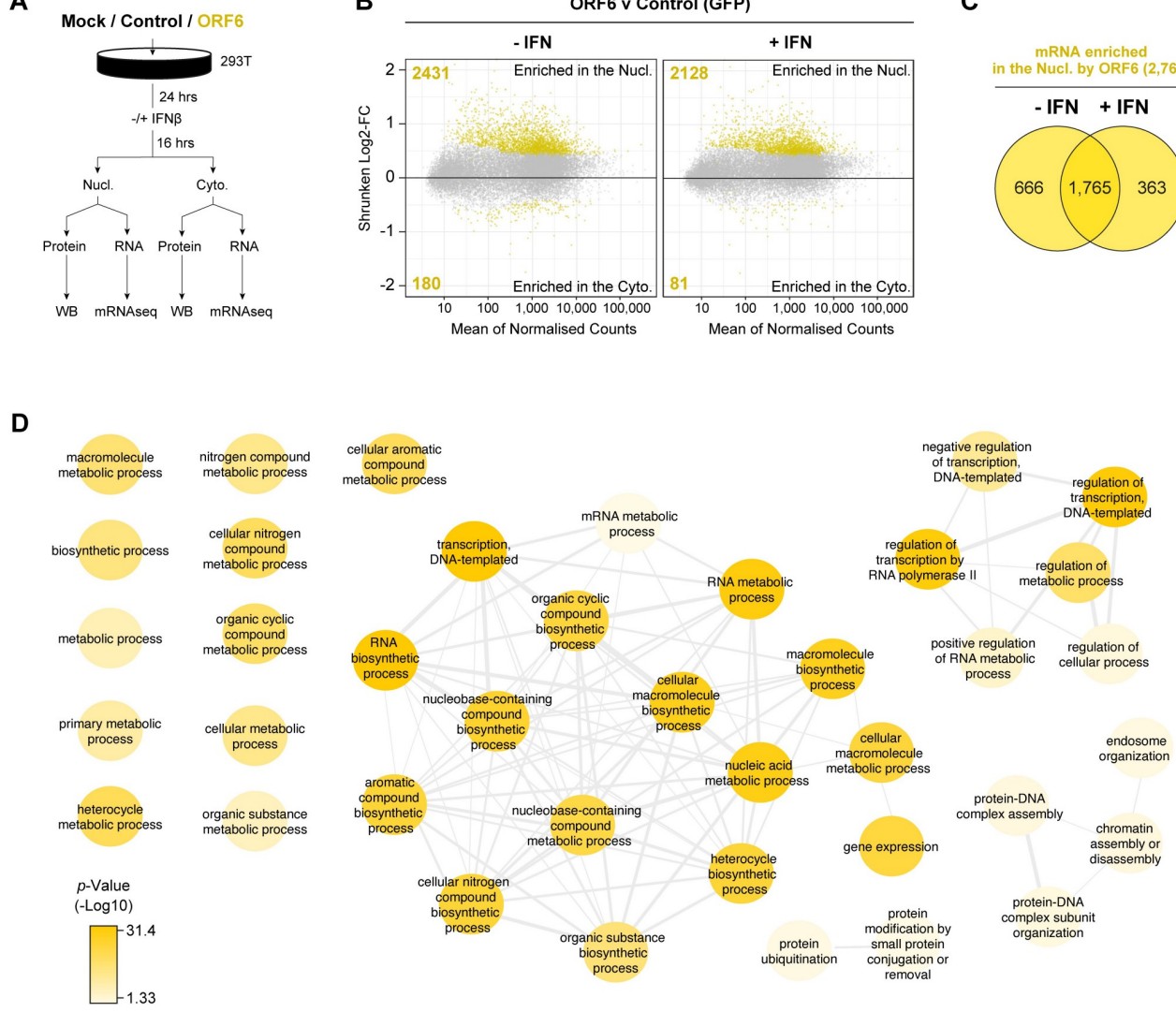

**Fig 3. ORF6 inhibits the nuclear export of cellular mRNA.** HEK293T cells were transfected with pLVX-EF1α-GFP or pLVX-EF1α-ORF6 or left untransfected (mock). After 24h, cells were either treated with IFN-β (1,000 units/ml) for 16h or left untreated. Cells were then harvested and fractionated into nuclear (Nucl) and cytoplasmic (Cyto) fractions. Cells were either lysed for immunoblot analysis (see S4A Fig) or RNA was extracted for mRNAseq. (A) Schematic of experiment design. (B) The log2-fold change (Log2-FC) in mRNA abundance was calculated between the Nucl and Cyto fractions for both GFP and ORF6 expressing cells, without and with IFN treatment (see S4B Fig). The log2-FC in mRNA abundance was then directly compared between ORF6 and GFP, without (left panel) and with (right panel) IFN, to determine which mRNAs were specifically enriched in the nucleus or cytoplasm by ORF6. The log-2FC was weighted against the adjusted *p*-value (shrunken Log2-FC) to show significantly enriched mRNAs (yellow points). The number of mRNAs significantly enriched is shown. The mRNA count was normalised and averaged between the three biological repeats. (C) Venn diagram of the number of mRNAs that were significantly enriched in the nucleus by ORF6 with and without IFN. (D) mRNAs significantly enriched by ORF6 without IFN treatment were analysed for enriched GO terms, which were filtered and mapped using the REVIGO tool. Highly similar GO terms are grouped and linked. Colour intensity reflects significance levels.

significantly upregulated in either the cytoplasm and/or nucleus after IFN-β treatment. While most upregulated genes were previously reported ISGs, some were not, therefore, to specifically refer to the genes upregulated in our study, we labelled them interferon upregulated genes (IUGs) (S5B Fig, S2 Table). Importantly, these IUGs encoded many proteins with roles in inhibiting viral replication and immune regulation (S5C Fig) and included well characterised ISGs like BST2 and IFITM1. Next, as ORF6 has been reported to inhibit IFN signalling, we measured mRNA expression in the presence of ORF6 compared to control cells in the

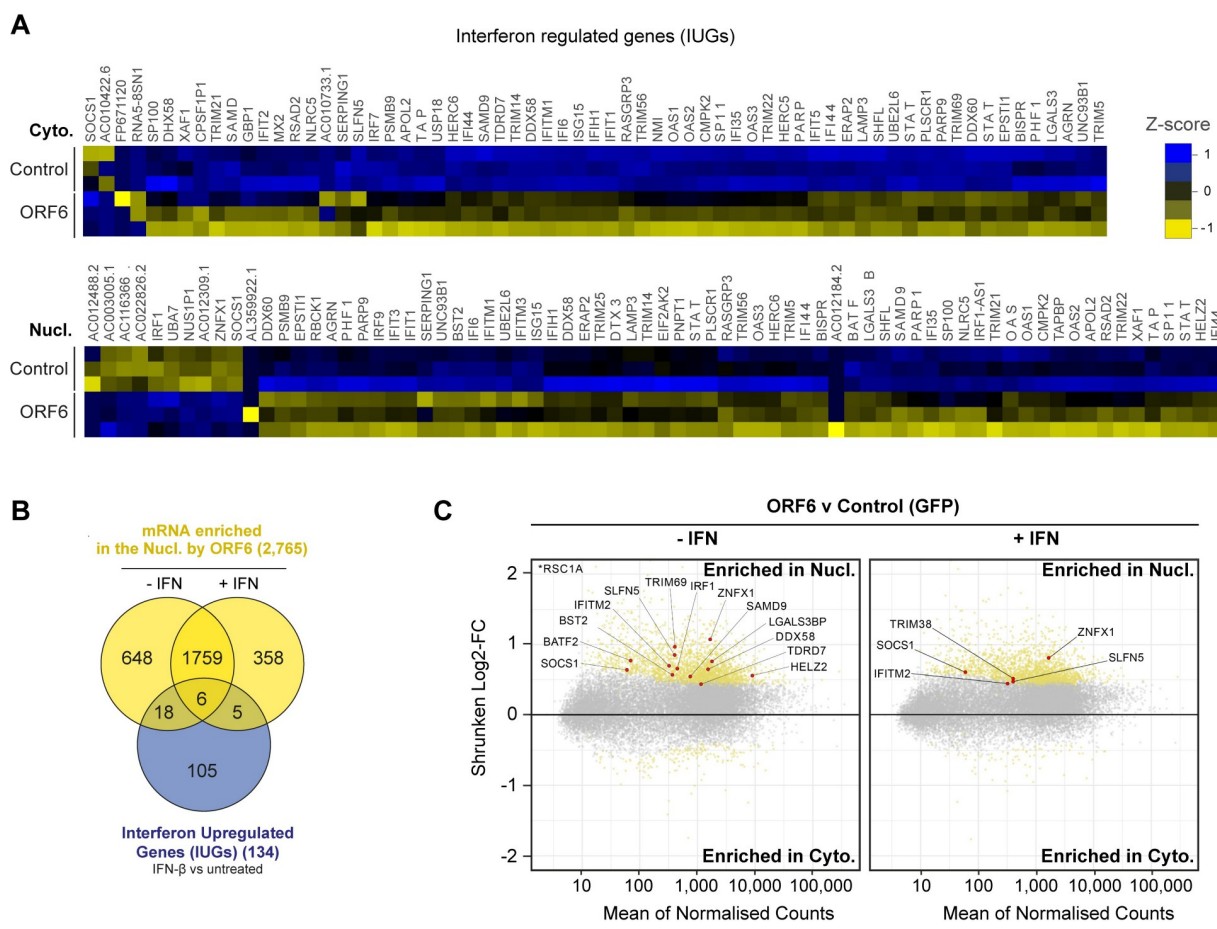

**Fig 4. ORF6 inhibits the nuclear export of IFN-upregulated mRNA.** (A) Interferon upregulated genes (IUGs) were identified as mRNAs that were significantly upregulated after IFN treatment compared to untreated cells (see S5A Fig). These IUGs were then compared to the list of mRNAs that were significantly upregulated or downregulated by ORF6 compared to the GFP control (see S5E Fig), in either the Cyto (upper panel) or Nucl (lower panel) fractions. The heat maps show the Z-score (scaled and centred per-gene expression) in IUG expression in cells expressing GFP (Control) or ORF6 after IFN treatment, for three biological repeats. (B) Venn diagram showing the number of IUG mRNAs that were significantly enriched in the nucleus by ORF6 with and without IFN treatment. (C) The log2-fold change (Log2-FC) in mRNA abundance was calculated between the Nucl and Cyto fractions for both GFP and ORF6 expressing cells, without and with IFN treatment, as in Fig 3B, to determine which mRNA were specifically enriched by ORF6. The log-2FC was weighted against the adjusted *p*-value (shrunken Log2-FC) to show significantly enriched mRNA (yellow points). IUGs are highlighted in red and labelled.

context of IFN-β treatment to determine if the expression of any of these identified IUGs was modulated by ORF6. Interestingly, of the 134 IUGs we identified, 69 IUGs were significantly downregulated in the cytoplasm in the presence of ORF6 (yellow tones) and 2 were significantly upregulated (dark blue tones) (Fig 4A, top panel). In the nuclear fraction, 62 IUGs were significantly downregulated in the presence of ORF6 while 10 were significantly upregulated (Fig 4A, bottom panel). In total, 79 unique IUGs were significantly downregulated by ORF6, while 11 unique IUGs were upregulated (S5D Fig, S3 Table). This supports the notion that ORF6 can inhibit IUG mRNA expression. However, in addition to the IUGs, ORF6 modulated the expression of many other mRNA compared to the control. The numbers of mRNA species upregulated or downregulated in the nucleus and/or cytoplasm by ORF6 in IFN treated and untreated cells are summarised in S5E Fig and S4 Table.

Next, we further investigated the effect of ORF6 on the IUGs we had identified. Intriguingly, of the 2,431 mRNA species that were enriched in the nucleus by ORF6 in the absence of

IFN treatment (Fig 3B), 24 were identified as IUGs (Fig 4B), including the antiviral factors TRIM69, BST2, ZNFX1, IRF1, and DDX58 (RIG-I) (Fig 4C, left panel). This suggests that ORF6 blocks the mRNA export of IUGs even when they are expressed at a basal level, which has significant implications for the antiviral state of the host cell. Furthermore, after IFN stimulation, ORF6 enriched 11 IUGs in the nucleus including TRIM38, SOCS1, IFITM2, SLFN5 and ZNFX1 (Fig 4C, right panel). Six of the IUGs were enriched in the nucleus in both IFN-treated and untreated cells (Fig 4B). Altogether, we identified 29 IUGs that were retained in the nucleus in the presence of ORF6 (Fig 4B, S3 Table) suggesting that in addition to inhibiting ISG expression, ORF6 can disrupt innate signalling by preventing the nuclear export of mRNA species that encode key antiviral factors that are both basally expressed and induced by IFN-β, like RIG-I, IRF1 and ZNFX1.

## SARS-CoV-2 inhibits the export of cellular mRNA

Our results demonstrate that expression of ORF6 in cells inhibits mRNA nuclear export with the effect of inhibiting protein expression. However, in the context of a viral infection, another viral protein may counteract the actions of ORF6. Therefore, to investigate whether mRNA export is also blocked during a SARS-CoV-2 infection, we analysed the nuclear accumulation of specific mRNAs in infected cells. Vero cells were infected with one of four SARS-CoV-2 variants: Eng/2, Alpha, Beta or Delta at a high MOI for 24h, after which cells were either fixed, permeabilised and probed with an anti-ORF6 antibody or separated into Nucl and Cyto fractions and processed for protein and RNA extraction. For all infections, ORF6 predominantly localised to the cytoplasm with some perinuclear staining (Fig 5A). The pattern of ORF6 expression was less diffuse in SARS-CoV-2(delta) infected cells than the other infections. We also saw some co-localisation of ORF6 with Nup98 (S6A Fig). Immunoblot analysis of cell lysates for HSP90 (cytoplasmic) and Histone H3 (nuclear) confirmed that fractionation of samples was successful (S6B Fig). ORF6 was also detected predominantly in the cytoplasm for most infections, although in two independent infections with SARS-CoV-2(delta) there was more ORF6 in the nuclear fraction (S6B and S6D Fig). This was probably linked to the poor viability of SARS-CoV-2(delta)-infected cells. In order to investigate mRNA export, we chose eight genes to analyse from our mRNAseq data: We identified three mRNAs that were most enriched in the nucleus in the presence of ORF6 (FAM222A, SPOCK1, SERTAD1, together classed "non-antiviral") (S6E Fig, red dots) as well as four mRNA encoding antiviral IUGs that were significantly enriched in the nucleus (RIG-I, IRF1, ZNFX1 and TRIM38) (Fig 4B, red dots). GAPDH was not significantly enriched in the nucleus with ORF6 and so was used as a control. Overall, all four SARS-CoV-2 variants increased the average Nucl:Cyto ratio of FAM222A, SPOCK1 and SERTAD1 mRNAs compared to uninfected cells (Fig 5B) while the Nucl:Cyto ratio of GAPDH mRNA remained constant. This suggests that export of these mRNA species is blocked in the context of SARS-CoV-2 infection. Strikingly, there was a large increase in the average Nucl:Cyto ratio of both RIG-I and IRF1 mRNA after infection with all four variants (Fig 5B). ZNFX1 was also enriched in the nucleus for all infections except SARS-CoV-2 (delta). However, TRIM38, which was only enriched in the nucleus by ORF6 after IFN treatment (Fig 4B, right panel), was only marginally enriched in the nucleus after infection. As Vero cells are unable to make IFN, we decided to repeat our infections in another cell line that can produce IFN in response to the infection. We chose A549 cells that express ACE2 and TMPRSS2 (A549-ACE2/TMPRSS2) and infected them with either SARS-CoV-2(alpha) or SARS-CoV-2(delta) for 24h. Cells were then either fixed, permeabilised and probed with an anti-ORF6 antibody or separated into Nucl and Cyto fractions for RNA extraction. In A549-ACE2/TMPRSS2 cells, ORF6 predominantly localised to the cytoplasm as in Vero cells

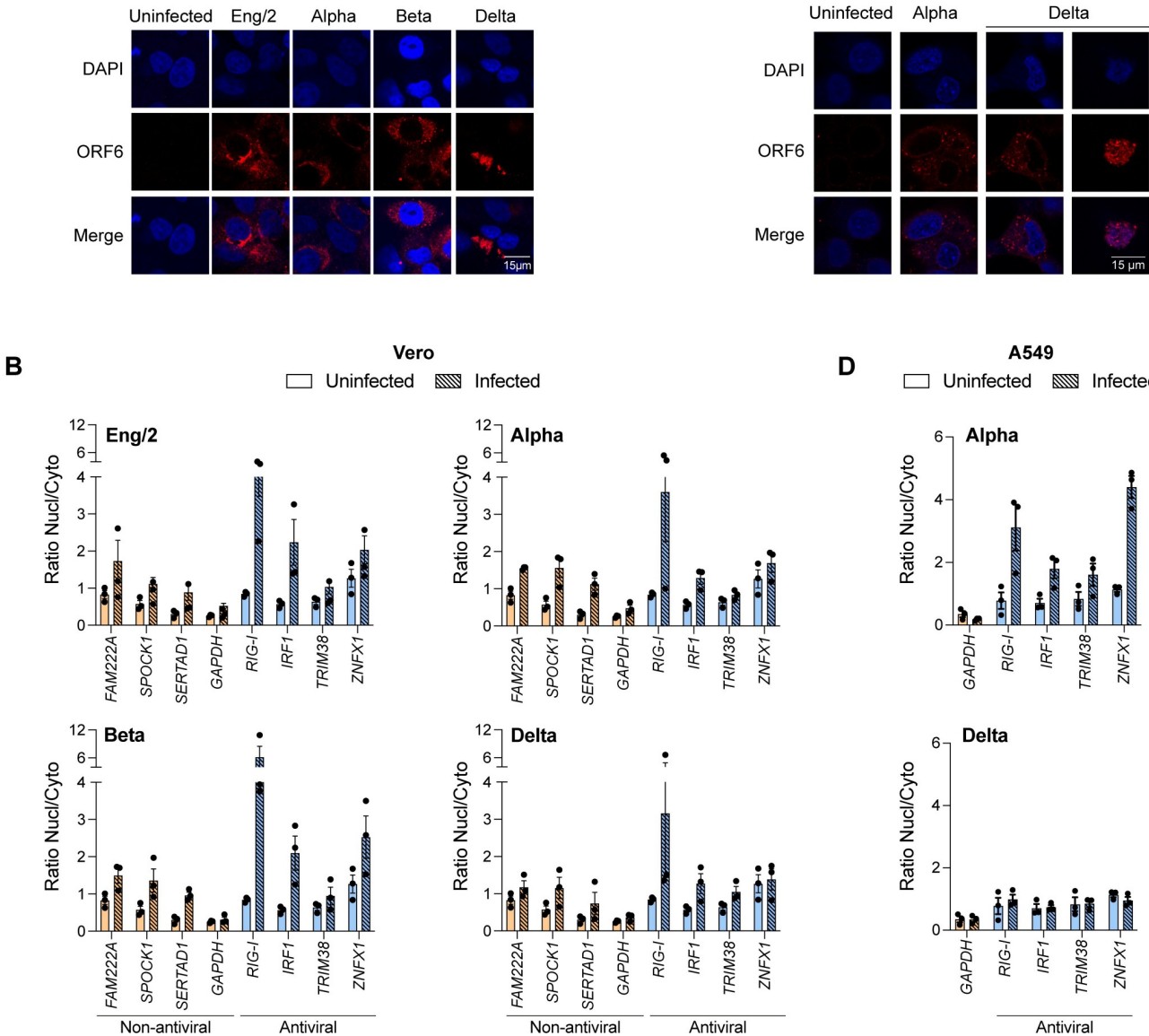

**Fig 5. SARS-CoV-2 inhibits the export of cellular mRNA.** (A, B) Vero cells were inoculated with four SARS-CoV-2 variants (Eng/2, Alpha, Beta, Delta) at a MOI ≥ 1 or left uninfected for 24h. (A) Cells were fixed and analysed by immunofluorescence with anti-ORF6 and DAPI. (B) Cells were fractionated, RNA was extracted, and cDNA was generated. mRNA levels of four non-antiviral (orange) and four antiviral (blue) proteins in both Nucl and Cyto fractions were measured by qPCR. The ratio of Nucl to Cyto mRNA was calculated and plotted. (C, D) A549-ACE2/TMPRSS2 cells were inoculated with two SARS-CoV-2 variants (Alpha, Delta) at a MOI ≥ 1 or left uninfected for 24h. (C) Cells were fixed and analysed by immunofluorescence with anti-ORF6 and DAPI. (D) Cells were fractionated, RNA was extracted, and cDNA was generated. mRNA levels of GAPDH (orange) and four antiviral (blue) proteins in both Nucl and Cyto fractions were measured by qPCR. The ratio of Nucl to Cyto mRNA was calculated and plotted. Points indicate individual biological repeats and error bars show the mean ± SEM.

(Fig 5C). As we observed before, SARS-CoV-2(delta) severely affect cell viability and in a subset of cells the pattern of ORF6 expression was altered, becoming predominantly nuclear. As the ORF6 sequence is the same for alpha and delta variants, the effect on cell viability is likely due to another viral protein(s) but at this point we cannot say which. Interestingly, we see a stronger reduction of nuclear mRNA export following SARS-CoV-2(alpha) infection in A549-ACE2/TMPRSS2 cells compared to Vero cells (Fig 5D, top panel), which may be due to

increased ISG induction in these cells. We could detect little effect on mRNA export in SARS-CoV-2(delta) infections (Fig 5D, bottom panel) but this was probably because the increased cell death limited the amount of RNA sample available for analysis.

Overall, we observed that SARS-CoV-2 infection induced retention of mRNA in the nucleus similar to ORF6 expression alone and that export of mRNAs involved in transcription (SERTAD1) and immune regulation (RIG-I and IRF1) was inhibited. Although other viral proteins may contribute to this phenotype, our data suggest that ORF6 could have a broad effect on cellular pathways during infection.

## Discussion

Coronaviruses encode several accessory proteins which are dispensable for *in vitro* replication but are presumably important for replication *in vivo* and disease pathogenicity [10]. This could make them good therapeutic targets. These proteins are not well characterised for SARS-CoV-2, so we decided to investigate their function. To do this, we generated a panel of HIV-1 VLPs encoding each individual accessory protein with which to introduce each protein into various cells for defined functional assays. Interestingly however, we were repeatedly unable to generate VLPs encoding ORF6 (Fig 1A). We therefore determined how ORF6 inhibits VLP production. We showed that ORF6 not only supresses expression of the HIV-1 structural protein Gag, which is required for VLP synthesis, but it also reduced MLV Gag (S1 Fig) and mApple expression (Fig 1D), implying that ORF6 blocks protein expression more broadly, in a sequence-independent manner.

Protein expression could have been inhibited via multiple mechanisms, however, ORF6 has been reported to interact with Rae1 and Nup98 [13,14,25] which are involved in nuclear import and export suggesting that ORF6 may modulate mRNA nuclear export resulting in decreased protein expression. We confirmed that ORF6 from both SARS-CoV-2 and SARS-CoV-1 interacted with Rae1, and that this interaction was important for the block to protein expression as mutants that were unable to interact also failed to inhibit protein expression (Figs 1F and 2). Furthermore, we showed directly that ORF6 did indeed inhibit mRNA export (Figs 3, 4 and S3). Interestingly, the herpesvirus ORF10 protein and the VSV M protein also interact with the same region of Rae1 [20,22,28] and block mRNA nuclear export. VSV M broadly inhibits the loading of cellular mRNA to the Rae1-Nup98 complex, while herpesvirus ORF10 can still form a complex with cellular mRNAs. Here, ORF6 only interacted with Nup98 indirectly through Rae1, similar to herpesvirus ORF10. ORF6 has been reported to dislocate Rae1 and Nup98 from the NPC, but whether this mislocalisation causes the mRNA export block or is a consequence of mRNA export disruption is not known [25]. A very recent paper suggests that although overexpressed ORF6 is localised to membranes, it is only a peripheral membrane protein (not transmembrane), and that ORF6 mis-localised proteins were still able to block nuclear trafficking [29]. Furthermore, in a recent preprint, Yoo *et al.* also report that membrane localisation is dispensable for nuclear transport [30]. Indeed, much of ORF6 is found associated with membranes other than the nuclear envelope where Nup98 resides, perhaps suggesting that ORF6 has other, uncharacterised functions. Yoo *et al.* propose that oligomerization of ORF6 may cross-link the FG domains of Nup98, obstructing the nuclear pore and preventing transport in either direction [30]. However, as our studies show that the ORF6 interaction with Nup98 is indirect, any cross-linking would have to be via Rae1 binding. Thus, we think it more likely that ORF6 inhibits nuclear export via preventing RNA accessing the pore, as suggested in a recent publication of the structure of ORF6 interacting with the Rae1-Nup98 complex [31].

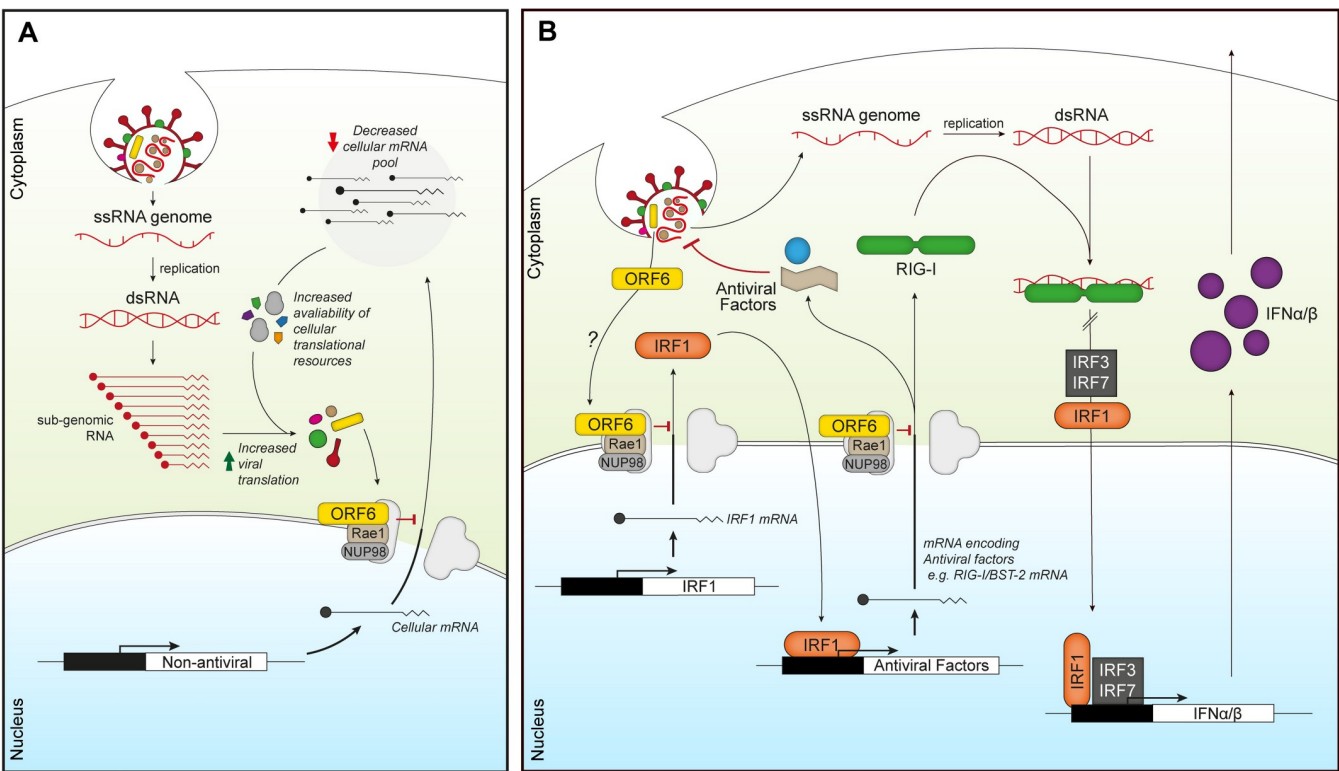

**Fig 6. ORF6 inhibits mRNA export to favour viral translation and supress innate signalling.** (A) Incoming or expressed ORF6 (yellow rectangles) binds to the Rae1-Nup98 complex (grey/brown) blocking the export of cellular mRNA (black lines). This subsequently reduces the cellular mRNA pool, increasing the availability of cellular translational machinery for viral translation as well as decreasing the expression of cellular proteins. (B) Incoming or expressed ORF6 (yellow) binds to the Rae1-Nup98 complex (grey/brown), inhibiting export of cellular mRNA encoding IRF1 (orange). This prevents the translation of IRF1, blocking IRF1 regulation of the transcription of additional steady state antiviral factors, like RIG-I and BST-2. ORF6 also inhibits nuclear export of mRNA encoding RIG-I (green), preventing detection of viral dsRNA produced during coronavirus replication. This helps reduce IRF1/3/7 activity and subsequent transcription of IFNα/β (purple), which prevents IFNα/β inducing an antiviral state in an autocrine and paracrine manner.

Blocking cellular mRNA nuclear export, thereby reducing translation of cellular proteins, would be advantageous for viral replication as SARS-CoV-2 replicates in the cytoplasm and would therefore be able to exploit the limited cellular translational machinery in favour of viral translation (Fig 6A). In addition, by blocking mRNA export, ORF6 can also affect other cellular processes beyond export, as inhibiting the translation of key cellular proteins will modulate their downstream processes. Excitingly, using mRNAseq, we have revealed that ORF6 can prevent the nuclear export of mRNAs encoding antiviral factors (Fig 4), both after IFN treatment, for example IFITM2 and ZNFX1, and more provocatively, in untreated cells. This implies that ORF6 can modulate the basal, steady state level of immunity in cells. Indeed, DDX58(RIG-I) and ZNFX1, which were enriched in the nucleus following both ORF6 expression in untreated cells (Fig 4) and during SARS-CoV-2 infection (Fig 5), are key pattern recognition receptors (PRRs) that detect viral dsRNA [32,33]. ZNFX1 has been postulated to be involved in the very early stages of IFN signalling, as one of the earliest viral sensors, due to its higher constitutive expression compared to both MDA5 and RIG-I [32]. Downregulation of these sensors would prevent the induction of an antiviral state and allow SARS-CoV-2 to replicate undetected (Fig 6B). The constitutively expressed transcription factor, IRF1, which helps regulate basal expression of antiviral factors like BST2 and RIG-I [34,35] was also targeted by ORF6. IRF1 mRNA is dependent on Nup98 and Rae1 for export and its corresponding protein has a relatively short half-life of around 20–40 mins [23,36,37]. Inhibiting IRF1 mRNA export would therefore have

major consequences for the induction of innate immunity [23,38,39]. Interestingly, ORF6 from SARS-CoV-1 is thought to be packaged into virions [40] and, thus, the protein is present in the cell as soon as virions infect. Incoming ORF6 could therefore downregulate PRRs before viral replication generates dsRNA intermediates, impeding the earliest antiviral warning system whilst also restricting the expression of further antiviral factors that should induce an antiviral state in the cell [40,41]. It would be interesting to know if SARS-CoV-2 ORF6 is also packaged into virions.

ORF6 from SARS-CoV-2 and SARS-CoV-1 have been shown to inhibit ISG and type-I IFN expression by preventing STAT1/2 and IRF3 nuclear translocation [12,14,21,42,43]. Another recent study used immunofluorescence to show that mRNA accumulated in the nucleus of ORF6 expressing cells [13]. Here, we show that ORF6 alone, or a viral infection, can inhibit nuclear export of a broad range of mRNA, including mRNA of antiviral proteins and transcription factors, which could have a downstream effect on ISG expression. Indeed, IRF1 regulates the expression of a range of ISGs by acting as a positive regulator of the JAK/STAT signalling pathway (Fig 6) [44]. Although ORF6 clearly inhibits mRNA nuclear export, the multi-functional SARS-CoV-2 protein, NSP1, has also recently been reported to inhibit mRNA export by blocking NXF1 docking to the NPC [24]. This redundancy in targeting mRNA export suggests that blocking this pathway gives the virus a clear survival advantage [45]. Further work, potentially knocking out ORF6 or NSP1 proteins individually, will investigate how ORF6 and NSP1 complement each other and whether this can be exploited in new therapeutic strategies directed against Sarbecoviruses [24,46].

## Materials and methods

### Cell lines

HEK293T, HeLa, Mus dunni tail fibroblast (MDTF), A549 pLV HYGRO-ACE2 NEO-TMPRSS2 (known as A549-ACE2/TMPRSS2) and Vero cell lines were maintained in Dulbecco's modified Eagle medium (DMEM, Thermo Fisher), supplemented with 10% heat-inactivated foetal bovine serum (FBS, Biosera) and 1% Penicillin/Streptomycin (Sigma). A549-ACE2/TMPRSS2 cells were further maintained in medium containing hygromycin (200 µg/ml) and G418 (2 mg/ml). A549-ACE2/TMPRSS2 cells were kindly gifted from Suzannah Rihn from the Centre for Virus Research. Cells were grown in a humified incubator at 37˚C and 5% $CO_2$. Stock cells were tested for mycoplasma contamination and authenticated by short-tandem repeat (STR) profiling.

### Plasmids and site-directed mutagenesis

SARS-CoV-2 ORF accessory proteins were expressed in cells via transfection or transduction with retroviral VLPs. The plasmids used to generate HIV-1 and MLV VLPs for transduction of ORFs, pCMVΔ8.91 and pHIT60 respectively, and pVSV-G have been described before [47,48]. For expression of each SARS-CoV-2 ORF accessory protein, two different plasmids were used. For the initial experiments on Gag expression (Figs 1A–1C, S1A and S1B) and ORF6 co-expression with Rae1 (Fig 2A), a pLVX-IRES-Puro vector encoding an individual ORF was used. The ORF genes were initially synthesised by GeneArt (Thermo Fisher) and first cloned into a pTriEX-6 plasmids using XmaI and SacI restriction sites. The pLVX-Strep-pII-ORF-IRES-Puro plasmids were then cloned using the NEBuilder HiFi DNA Assembly Cloning Kit (NEB) from the pTriEX-6 plasmids, using the primers listed in Table 1. For the remaining assays that required ORF detection or immunoprecipitation, a pLVX-EF1α-SARS-CoV-2-ORF-2xStrep/2xStrep-ORF-IRES-Puro encoding one of the nine ORFs or pLVX-EF1α-eGFP-2xStrep-IRES-Puro plasmid was used, referred to as twin-strep-tag. These

**Table 1. Sequences of Primers used for cloning.**

| Mutation/Modification | Primer | Sequence (5′-3′) |
|---|---|---|
| E55A | FWD | cagttggacgaggCgcagcctatgg |
| | REV | ccataggctgcGcctcgtccaactg |
| M58A | FWD | gaggagcagcctGCggaaatagacctc |
| | REV | gaggtctatttccGCaggctgctcctc |
| E59A | FWD | gagcagcctatggCaatagacctcgaag |
| | REV | cttcgaggtctattGccataggctgctc |
| D61A | FWD | ctatggaaatagCcctcgaaggcgg |
| | REV | ccgccttcgaggGctatttccatag |
| R305G | FWD | gacaaagatgccGgaacaaaactaaaaac |
| | REV | gttttttagttttgttcCggcatctttgtc |
| T306I | FWD | caaagatgccagaaTaaaactaaaaac |
| | REV | gttttttagttttAttctggcatctttg |
| RT305/306GI | FWD | gacaaagatgccGgaaTaaaactaaaaac |
| | REV | gttttttagttttAttcCggcatctttgtc |
| HA-RAE1 | FWD | caggttcaaaatgTACCCATACGATGTTCCAGATTACGCTagcctgtttggaac |
| | REV | gttccaaacaggctAGCGTAATCTGGAACATCGTATGGGTAcattttgaacctg |
| Nup98-MYC | FWD | gagcccagaggtgGAACAGAAACTGATCTCTGAAGAAGACCTGtaggaattctgcag |
| | REV | ctgcagaattcctaCAGGTCTTCTTCAGAGATCAGTTTCTGTTCcacctctgggctc |
| pTriEX-6 | FWD | tagaggatctatttccggtgaattcACCATGGCAAGCTGGAGC |
| | REV | ggggggagggagaggggcgggatccATGAGCGGAACTACCGCG |

plasmids were kindly gifted from Nevan Krogan (Addgene plasmid #141383–4, #141387–90, #141392–95); http://n2t.net/addgene:141383-4, 141387–90, 141392–95; RRID: Addgene_141383–4, 141387–90, 141392–95) [5]. To generate the pLVX-EF1α-CoV-1-ORF6-2xStrep-IRES-Puro plasmid, the ORF6(CoV-1)-2XStrep gene was synthesised by GeneArt and cloned in using the EcoRI and BamHI restriction sites. ORF6 mutations were introduced into the pLVX-StrepII-SARS-CoV-2-ORF6-IRES-Puro and pLVX-EF1α-SARS-CoV-2-ORF6-2xStrep-IRES-Puro plasmids using the QuickChange II-XL site-directed mutagenesis kit (Agilent) according to the manufacturer's instructions using SDM primers listed in Table 1. The following plasmids were sourced from Addgene, psfGFP-N1 (Addgene, #54737) and pmApple-N1 (Addgene, #54567). pCMVsport6Rae1 (MGC:117333) was purchased from Horizon Discovery Biosciences Limited. Nup98 was amplified from a HeLa cDNA library from Clontech/Takara. Affinity tags were introduced by NEBuilder HiFi DNA Assembly Cloning Kit (NEB) into the expression vectors encoding Rae1 and Nup98 using primers in Table 1. Sequences were verified by Sanger sequencing (Source Bioscience).

## Virus-like particles (VLP) production

HIV-1 and MLV virus-like particles were generated by co-transfecting HEK293T cells with plasmids, pCMVΔ8.91 (HIV-1) or pHIT60 (MLV) and pVSV-G using lipofectamine 2000 (11668019, Invitrogen). For generating GFP reporter VLPs, pCSGW (HIV-1) or pczCFG2fEGFPf (MLV) were also added [48,49]. The plasmids were used at a ratio of 1:1:1 for generation of MLV VLPs and at a ratio of 10:10:1 for HIV-1 VLPs. After 16h, cells were treated with 10 mM sodium butyrate for 8h, and supernatant containing VLPs harvested 24h later. To measure viral titres, HeLa cells were transduced with serially diluted supernatant containing the VLPs. After 72h, cells were selected with media containing puromycin (1 μg/ml) for 10

days, after which cells were washed, fixed, and stained with methanol containing crystal violet. Colonies were counted and viral titre calculated.

## Immunoblotting

Cells were lysed in ice cold radioimmunoprecipitation assay (RIPA) buffer (Thermo Fisher) or IP buffer (50 mM Tris-HCl, pH 7.4 at 4˚C, 150 mM NaCl, 1 mM EDTA supplemented with 0.5% Nonidet P40) containing protease inhibitor cocktail (Roche) and DNAse (#88701, Thermo Fisher). Protein lysates were boiled at 95˚C for 5 mins in 6x SDS sample buffer (Thermo Scientific). Proteins were separated on 4–12% Bis-Tris SDS-PAGE gels (Thermo Fisher) or 4–20% mini-PROTEAN TGX (Bio-Rad) and transferred to a PVDF membrane. Primary antibodies used were: anti-HIV-1 p24 (ARP432, CFAR), anti-GFP (sc-9996, SCBT), anti-HA (C29F4, CST), anti-Strep-tag (34850, Qiagen), anti-HSP90 (4874, CST), anti-Histone H3 (14269, CST), anti-alpha-Tubulin (VMA00051, BIO-RAD), anti-Lamin B1 (66095-1-Ig, Proteintech), anti-Nup98 (C39A3, CST), anti-Rae1 (ab124783, Abcam), anti-pSTAT(Y701) (D4A7, CST), anti-ORF6/9b (MRC PPU, Dundee University). Secondary antibodies include, anti-mouse (61–6520, Thermo Fisher), anti-sheep (ab6747, Abcam) and anti-rabbit (31460, Thermo Fisher) HRP-conjugated; anti-mouse IRDye 680RD/800CW (926-68072/926-32212, Li-Cor) and anti-rabbit IRDye 700CW/800CW (926-68073/926-32213, Li-Cor). Blots were imaged on a Chemidoc MP imaging system (Bio-Rad) or Odyssey Clx imaging system (LICOR). Band intensity was quantified from the ImageLab files using the ImageLab 6.1 software and Gag band intensities were normalised to HSP90 band intensities. Band intensity was quantified from the ImageLab files using the ImageLab 6.1 software and Gag band intensities were normalised to HSP90 band intensities.

## Immunoprecipitation

HEK293T cells were grown in 10 cm$^2$ culture plates transfected and incubated for 16h after which media was replaced. After a further 8h, cells were harvested in Pierce IP lysis buffer (Thermo Fisher), supplemented with protease inhibitor cocktail and Pierce DNAse (Thermo Fisher). Twin-strep fusion proteins were immunoprecipitated with MagStrep "type3" XT beads (IBA) as per the manufacturer's instructions. Proteins were eluted with biotin (50 mM) and boiled in 6x SDS sample buffer.

## Subcellular fractionation

HEK293T cells were transfected with psfGFP-N1 and either treated with LMB (Merck) at 5nM; or co-transfected with plasmids expressing pLVX-EF1α-ORF6, ORF6(M58A) or ORF9b [5]. At the indicated times post-transfection, 1x10$^6$ cells were harvested and processed with the PARIS kit (Life Technologies) following the manufacturer's instructions. Parallel, samples were lysed as total cell lysates or were fractionated into cytoplasm and nuclear fractions. Each sample was then split in two: half was used for protein analysis and half was used for RNA extraction. Protein content in the total cell lysate and in the fractions was measured by BCA assay (Thermo Fisher) using a FLUOstar Omega plate reader (BMG Labtech). An equivalent amount of each fraction compared to the total cell lysate was analysed by immunoblotting.

## Quantitative PCR

To remove possible DNA contamination, extracted RNA from the nuclear and cytoplasmic fractions was treated with DNase using the DNA-free kit (Invitrogen) and following the manufacturer's instructions. Then, 100ng of RNA was converted to cDNA for 1h at 37˚C using the

Omniscript RT kit (Qiagen) and random primers (Promega). Quantitative PCR (qPCR) analysis was performed in TaqMan Fast Advanced master mix (Thermo Fisher) with 900nM primers and 250nM probes, or 1X Taqman gene expression assays. The reactions were performed on a 7500 fast real-time PCR system (Applied Biosystems) using standard cycling conditions: 50˚C for 2 min, 95˚C for 10 min followed by 40 cycles of 95˚C for 15s and 60˚C for 1min. To calculate DNA copy numbers, standard curves were generated from serial dilutions of the indicated cDNA in water. The following primers and probes were used; sfGFP: *for* 5'-GCGCACC ATCAGCTTCAAGG, *rev* 5'-GTGTCGCCCTCGAACTTCAC and *probe* 5'-*FAM*-CGGCAC CTACAAGACCCGCGC-*TAMRA*.

## mRNAseq

The quality of extracted RNA was checked by Agilent TapeStation 42000 on Agilent RNA ScreenTape (Agilent technologies) before proceeding to library preparation. Only samples with a RINe of 7 or higher were used for Library preparation. RNA-seq libraries were generated using the NEBNext Ultra Directional RNA Library Prep kit for Illumina with NEBNext Poly(A) mRNA Magnetic Isolation Module (both New England BioLabs). Samples were adjusted to 400 ng total RNA and spiked with diluted 1/100 ERCC Spike-In Mix (4456740, Invitrogen) and Drosophila melanogaster, embryo Poly A+ RNA, 5 µG (636224-Takara) using 1 µl from each. PCR Enrichment of Adaptor Ligated DNA was run with 12 cycles and SPRI beads (Beckman) were used for library clean up. Agilent DNA Screen tapes (Agilent Technologies) were used to check library size and quality. Samples were sequenced on an Illumina Novaseq 6000 platform with 180 pM loading concentration in single-end run using 100-bp reads. Enriched GO terms (Biological process) were filtered and mapped using the REViGO tool available at http://revigo.irb.hr/ [50]. The raw RNAseq data from this study can be accessed from ENA via accession PRJEB49943.

## mRNAseq data analysis

A custom genome joining the Human (Ensembl GRCh38.100) and D. melanogaster (Ensembl BDGP6.104) genomes was prepared, and reads were trimmed, aligned, and counts produced using the nf-core rnaseq pipeline [51]. Normalisation and differential expression analyses were conducted within DESeq2 [52]: normalisation factors for gene-level read counts were produced using counts for the D. melanogaster spike-in controls and differential gene expression calculations were subsequently performed for Human genes alone using the pre-calculated size factors. Independent hypothesis weighting was conducted to optimise the power of p value filtering using the IHW package [53] and Log2 fold-change shrinkage was performed using ashr package [54]. Result tables were subsequently filtered on (IHW-adjusted) q<0.05.

## SARS-CoV-2 infections

The following viruses were used in the infections: The Wuhan-like reference isolate (referred to as the Eng/2) was the hCoV-19/England/02/2020, obtained from the Respiratory Virus Unit, Public Health England, UK (GISAID EpiCov accession EPI_ISL_407073). The B.1.1.7 (Alpha) isolate was the hCoV-19/England/204690005/2020, which carries the D614G, Δ69–70, Δ144, N501Y, A570D, P681H, T716I, S982A, and D1118H mutations [55], obtained from Public Health England (PHE), UK, through Prof. Wendy Barclay, Imperial College London, London, UK. The B.1.351 (Beta) virus isolate was the 501Y.V2.HV001, which carries the D614G, L18F,D80A, D215G, Δ242–244, K417N, E484K, N501Y, A701V mutations, and was kindly provided by Prof. Alex Sigal and Prof. Tulio de Oliveira [56], sequencing of viral isolates received identified the Q677H and R682W mutations at the furin cleavage site in

approximately 50% of the genomes, which was maintained upon passage in cell culture. The B.1.617.2 (Delta) isolate was MS066352H (GISAID accession number EPI_ISL_1731019), which carries the T19R, K77R, G142D, Δ156-157/R158G, A222V, L452R, T478K, D614G, P681R, D950N, and was kindly provided by Prof. Wendy Barclay, Imperial College London, London, UK through the Genotype-to-Phenotype National Virology Consortium (G2P-UK). Viruses were titrated by plaque assay on Vero E6 cells. Serial ten-fold dilutions of virus are incubated for 40 mins at RT on a well of confluent Vero E6 cells. The virus inoculum is then removed, and the cells overlayed with a semisolid overlay containing 1.2% Avicel in MEM. Cells are incubated for two days at 37˚C, 5% $CO_2$ after which the overlay is removed. Cells were fixed and stained with 4% paraformaldehyde containing 0.2% toluidine blue for at least 30 mins. For the infections, Vero E6 or A549-ACE2/TMPRSS2 cells were seeded 24h before use to achieve 80–90% confluency. Cells were washed, medium was removed from each well and 1 ml of serum-free DMEM added. The 1 ml of serum-free DMEM was removed and 500 μl virus diluted in virus growth medium was added to achieve an MOI of ≥1. Plates were incubated for approx. 30 mins at RT and then 1 ml virus growth medium was added to each well. Plates were incubated for 24h before RNA isolation and qPCR as described above using Taqman gene expression assays (Thermo Fisher) as follows: FAM222A (Hs00757936_m1), SPOCK1 (Hs0270274_m1), SERTAD1 (Hs00203547_m1), ZNFX1 (Hs01105231_m1), TRIM38 (Hs00197164_m1), RIG-I (Hs01061433_m1), GAPDH (Hs02786624_g1) and IRF1 (Hs00971964_g1).

## Immunofluorescence

HEK293T, Vero or A549-ACE2/TMPRSS2 cells were grown on glass coverslips and transfected or infected with SARS-CoV-2. Cells were fixed in 4% PFA in PBS for 10 mins at RT. Cells were then permeabilised with 0.1% triton in PBS and blocked in 3% BSA in PBS. Cells were labelled with the following primary antibodies, anti-Nup98 or anti-ORF6 diluted in 4% BSA in PBS for 1h at RT. Cells were then incubated with the following secondary antibodies, donkey anti-rabbit-AF568 (ab175692, Abcam) and donkey anti-sheep AF568 (A-21099, Invitrogen) diluted in 4% BSA in PBS for 1h at RT. Coverslips were washed and mounted on glass coverslips with ProLong Gold Antifade Mountant with DAPI (Thermo Fisher) and imaged with the Leica SP5 microscope and 100X 1.3NA oil immersion objective (Leica).

## Statistics

Statistical analysis was performed using GraphPad Prism 9 software. ($^*$, $P < 0.05$; $^{**}$, $P < 0.01$; $^{***}$, $P < 0.001$; $^{****}$, $P < 0.0001$, ns–not significant).

## Supporting information

**S1 Fig. SARS-CoV-2 ORF6 Inhibits MLV Gag expression.** (A, B) To generate MLV VLPs, HEK293T cells were transfected with plasmids encoding Gag-Pol(MLV), VSV-G and either increasing amounts of pLVX-StrepII-ORF6 or pLVX-StrepII-ORF9b as well as a GFP reporter plasmid, pczCFG2fEGFPf. As a control, VLPs were generated with an empty pLVX vector. (A) Transfected cells were lysed and analysed by SDS-PAGE and immunoblotting for Gag(MLV), ORF6 or ORF9b and HSP90. (B) MDTF cells were infected with increasing amounts of MLV VLPs. Three days post infection, the percentage of GFP positive cells was determined by flow cytometry. Graph shows the mean and range of two biological repeats. (C) To generate HIV-1 VLPs, HEK293T cells were transfected with plasmids encoding Gag-Pol(HIV-1), VSV-G, a GFP reporter plasmid, pCSGW and pLVX-EF1α encoding either twin-strep tagged ORF6, ORF9b or as a control an empty retroviral plasmid. HeLa cells were infected with HIV-1 VLPs.

Three days post infection, the percentage of GFP positive cells was determined by flow cytometry. From one biological repeat, cells were lysed and immunoblotted for anti-strep, Gag(HIV-1) and HSP90. Gag expression was quantified, normalised to HSP90, and shown in numbers below the Gag panel. (D) HEK293T cells were transfected with the pLVX-EF1α plasmid encoding either ORF6(CoV-1), ORF6(CoV-2) or the indicated ORF6 mutant as well as a plasmid encoding mApple. After 48h, cells were fixed and the mApple MFI measured by flow cytometry. The fold change (FC) in MFI is plotted relative to the control for independent biological repeats. Error bars show SEM. Transfected cell lysates were analysed by SDS-PAGE and immunoblotted with anti-strep.
(TIF)

**S2 Fig. ORF6 interacts with endogenous Rae1.** (A) HEK293T cells were transfected with plasmids encoding Twin-Strep-tagged GFP, ORF6(CoV-2) or ORF6(CoV-1) and HA-Rae1. After 24h, tagged proteins were immunoprecipitated with MagStrep beads and proteins eluted with biotin. Input lysates and eluate were separated by SDS-PAGE and analysed by immunoblotting for Strep, Nup98 and HA. (B) HEK293T cells were transfected with the pLVX-EF1α plasmid encoding Twin-Strep-tagged GFP or ORF6. After 24h, tagged proteins were immunoprecipitated with MagStrep beads and proteins eluted with biotin. Input lysates and eluate were separated by SDS-PAGE and immunoblotted for endogenous Rae1 and anti-Strep.
(TIF)

**S3 Fig. ORF6 inhibits GFP mRNA export in a Rae1-dependent manner.** (A-D) HEK293T cells were transfected with a plasmid expressing GFP or left untransfected (mock) and either treated with 5 nM Leptomycin B (LMB) or left untreated. After 16h, cells were either harvested as a total cell lysate (T) or fractionated into Nucl (N) and Cyto (C) fractions. Total cell lysates and cellular fractions were divided into two for either protein analysis (B) or RNA extraction (C,D). (A) Low magnification fluorescent images showing GFP expression in transfected cells before harvest. (B) Protein levels were quantified by BCA assay, proportional amounts of the fractions related to the total cell lysate were analysed by immunoblotting for GFP, HSP90 (cytoplasmic marker) and Histone H3 (nuclear marker). (C) RNA from Nucl and Cyto fractions was converted into cDNA and GFP mRNA copy numbers measured by qPCR. (D) The ratio of Nucl to Cyto GFP mRNA was calculated and plotted for individual biological repeats. Error bars show the mean ± SEM. (E-H) HEK293T cells were co-transfected with plasmids encoding either ORF6, ORF6(M58A) or ORF9b and GFP, or left untransfected (mock). After 32h, cells were either harvested as a total cell lysate (T) or fractionated into Nucl (N) and Cyto (C) fractions. Total cell lysates and cellular fractions were divided into two for either protein analysis (F) or RNA extraction (G,H). (E) Low magnification fluorescent images showing GFP expression in transfected cells before harvest. (F) Protein levels were quantified by BCA assay, proportional amounts of the fractions related to the total cell lysate were analysed by immunoblotting for Strep, GFP, HSP90 (cytoplasmic marker) and Histone H3 (nuclear marker). (G) RNA from Nucl and Cyto fractions was converted into cDNA and GFP mRNA copy numbers measured by qPCR. (H) The ratio of Nucl to Cyto GFP mRNA was calculated and plotted for individual biological repeats. Error bars show the mean ± SEM. Significance was calculated by one-way ANOVA with Turkey's post hoc test, $^*p<0.05$, $^{**}p<0.01$, ns–non-significant.
(TIF)

**S4 Fig. ORF6 inhibits the nuclear export of cellular mRNA.** (A) The cell fractionations used in Fig 3A, and B were analysed by immunoblotting with anti-Strep to label ORF6 and GFP, anti-pSTAT(Y701) to confirm activation of the type-I IFN signalling cascade and anti-HSP90 (cytoplasmic marker) or anti-Lamin B1 (nuclear marker) to confirm successful fractionation.

The analysis of three biological repeats is shown. (B) From the mRNAseq data in Fig 3. The log2-fold change (Log2-FC) in mRNA abundance was compared between the Nucl and Cyto fractions for both GFP and ORF6 expressing cells, without (upper panels) and with IFN treatment (lower panels). The log-2FC was weighted against the adjusted *p*-value (shrunken Log2-FC) to show mRNAs that are significantly enriched (GFP; black, ORF6; yellow) in either the cytoplasm or nucleus. The number of mRNAs significantly enriched are shown. The mRNA count was normalised and averaged between three biological repeats.
(TIF)

**S5 Fig. ORF6 inhibits the nuclear export of IFN-upregulated mRNA.** (A) From the mRNA-seq data in Fig 3. The Log2-FC in mRNA abundance was compared between cells treated with IFN or left untreated in both the Cyto (left panel) and Nucl (right panel) fractions. The log-2FC was weighted against the adjusted *p*-value (shrunken Log2-FC) to show mRNA species that are significantly upregulated or downregulated (blue). The number of mRNAs upregulated is shown. These were designated as Interferon upregulated genes (IUGs). (B) Venn diagram showing the number of IUGs significantly upregulated in Cyto. and Nucl. fractions. (C) GO ontology enrichment analysis of significantly upregulated IUGs from A. in both the Cyto and Nucl fractions. (D) Venn diagram showing the number of IUGs significantly downregulated (red) or upregulated (green) by ORF6 in Cyto. and Nucl. fractions, as described in Fig 4A. (E) From the mRNAseq data in Fig 3. The log2-fold change (Log2-FC) in mRNA abundance was compared between ORF6 and GFP expressing cells in the Cyto. and Nucl fractions, without (upper panel) and with IFN treatment (lower panel). This log2-FC was weighted against the adjusted *p*-value (shrunken Log2-FC) to identify significantly upregulated/downregulated mRNAs (in yellow), with the number of mRNAs shown. The mRNA count was normalised and averaged between the three biological repeats.
(TIF)

**S6 Fig. SARS-CoV-2 inhibits the export of cellular mRNA.** (A) Vero cells that were inoculated with SARS-CoV-2(beta) in Fig 5A were fixed, permeabilised and analysed by immunofluorescence with anti-Nup98, anti-ORF6 and DAPI. (B-D) Cell lysates from Fig 5B were analysed by immunoblotting for Strep to label ORF6, Nsp2, HSP90 (Cytoplasmic marker) and Histone H3 (Nuclear marker) to confirm successful fractionation. The analysis of three biological repeats are shown. (E) The plot shown in Fig 3B (right panel) highlighting the enrichment of the mRNA encoding the four non-antiviral proteins that were quantified by qPCR in Fig 5B.
(TIF)

**S1 Table. RNAseq analysis: Enriched mRNA species.**
(XLSX)

**S2 Table. RNAseq analysis: IUGs Identified.**
(XLSX)

**S3 Table. RNAseq analysis: Differentially expressed or enriched IUGs.**
(XLSX)

**S4 Table. RNAseq analysis: Differentially expressed mRNA species.**
(XLSX)

## Acknowledgments

We thank Nevan Krogan for gifting the twin-strep-tagged ORF protein plasmids via Addgene (Addgene plasmid #141383–4, #141387–90, #141392–95), Wendy Barclay, Prof. Alex Sigal and

Prof. Tulio de Oliveira for providing SARS-CoV-2 viral isolates and Suzannah Rihn for providing the A549-ACE2/TMPRSS2 cells. We are extremely grateful to Jimena Perez-Lloret, Robert Goldstone and Deb Jackson from the Crick ASF STP for running the RNAseq experiment. For the purpose of Open Access, the authors have applied a CC BY public copyright licence to any Author Accepted Manuscript version arising from this submission.

## Author Contributions

**Conceptualization:** Ross Hall, Anabel Guedán, Melvyn W. Yap, Jonathan P. Stoye, Kate N. Bishop.

**Data curation:** Ross Hall, Anabel Guedán, Melvyn W. Yap, George R. Young.

**Formal analysis:** Ross Hall, Anabel Guedán, Melvyn W. Yap, George R. Young, Kate N. Bishop.

**Funding acquisition:** Jonathan P. Stoye, Kate N. Bishop.

**Investigation:** Ross Hall, Anabel Guedán, Melvyn W. Yap, Ruth Harvey.

**Methodology:** Ross Hall, Anabel Guedán, Melvyn W. Yap, George R. Young, Ruth Harvey.

**Project administration:** Kate N. Bishop.

**Resources:** Ross Hall, Anabel Guedán, Melvyn W. Yap, Ruth Harvey, Jonathan P. Stoye, Kate N. Bishop.

**Supervision:** Kate N. Bishop.

**Writing – original draft:** Ross Hall, Anabel Guedán, Kate N. Bishop.

**Writing – review & editing:** Melvyn W. Yap, George R. Young, Ruth Harvey, Jonathan P. Stoye, Kate N. Bishop.

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
