## [Decision Letter · Decision Letter 0]

4 Apr 2022

Dear dr. Bishop,,

Thank you very much for submitting your manuscript "SARS-CoV-2 ORF6 disrupts innate immune signalling by inhibiting cellular mRNA export" for consideration at PLOS Pathogens. As with all papers reviewed by the journal, your manuscript was reviewed by members of the editorial board and by several independent reviewers. In light of the reviews (below this email), we would like to invite the resubmission of a significantly-revised version that takes into account the reviewers' comments.

Given the fact that the reviewers have major comments on the technical aspects (reviewer 1 and 3), all the major issues raised should be addressed if possible. In addition, the authors should discuss in detail recent publications, as indicated by reviewer 2.

We cannot make any decision about publication until we have seen the revised manuscript and your response to the reviewers' comments. Your revised manuscript is also likely to be sent to reviewers for further evaluation.

Sincerely,

Bart L. Haagmans

Guest Editor

PLOS Pathogens

Ron Fouchier

Section Editor

PLOS Pathogens

Kasturi Haldar

Editor-in-Chief

PLOS Pathogens

orcid.org/0000-0001-5065-158X

Michael Malim

Editor-in-Chief

PLOS Pathogens

orcid.org/0000-0002-7699-2064

Given the fact that the reviewers have major comments on the technical aspects (reviewer 1 and 2), all the major issues raised should be addressed if possible. In addition, the authors should discuss in detail recent publications, as indicated by reviewer 2.

Reviewer's Responses to Questions

**Part I - Summary**

Reviewer #1: In this work by Hall et al, the authors assess the affect of SARS-CoV-2 accessory proteins on host cells. The authors find that when attempting to make HIV VLPs, the co-expression of SARS2 accessory protein ORF6 blocks VLP production. Through a series of transfection and cell biology experiments, the authors show that ORF6 binds to Rea1 to inhibit protein expression. This interaction leads to the inhibition of cellular mRNA export from the nucleus, and is also due to the interaction with Nup98. The authors map the interaction domain to the C terminal tail of ORF6 demonstrating a key AA as E55, to its inhibitory function and the interactions for export block. Focusing on the nuclear export of RNA, the authors use fractionation and deep sequencing to identify mRNA that is sequestered in the nucleus when ORF6 is expressed and/or when cells are treated with IFN. This identified subsets of genes that are regulated by ORF6 expression, especially key ISGs including IRF1 and RIGI.

The paper can be divided into 2 parts, with the identification of ORF6 and its interaction with Nup98/Rae1 as the first part of the paper and the RNASeq fractionation as the second part of the paper. Much of the first part of the paper has already been published previously by other groups, with this work not adding different results to what is currently known of ORF6. The previous papers by Addetia et al, mBio 2021, Miorin et al 2020 PNAS, and Kato et al, 2021 BBRC. The previous work is only briefly mentioned in the introduction and only the Addetia et al paper mentioned in the discussion as it pertains to the mRNA localization in the nucleus, not the rest of their findings. The previous work should be referenced as finding the same as the first part of this manuscript and acknowledged for the same findings. The RNASeq and analysis of ORF6 localization is problematic for its interpretation of the data, differences in RNA levels seen as well as where ORF6 is in the cell.

Reviewer #2: The manuscript by Kate Bishop and coauthors explored the importance of ORF6 interacting with Rae1 disrupting innate immune response by inhibiting mRNA export.

Reviewer #3: The manuscript by Hall et al. explores the relevance of the SARS-CoV-2 accessory protein ORF6 in nucleo-cytoplasmic trafficking, which is a critical process in the cell response to environmental stress conditions, such as viral infections.

It was previously described that SARS-CoV-1 ORF6 protein disrupts nuclear import of STAT1 by retaining import factors (karyopherins A2 and B) at the ER/Golgi membrane (Frieman et al. J Virol. 2007;81(18):9812-24). SARS-CoV-2 ORF6 was also reported to directly interact with Nup98-Rae1 components of the nuclear pore complex (Gordon et al. Nature. 2020. 583, 459–468) and inhibit STAT1 and STAT2 nuclear translocation (Miorin et al. Proc Natl Acad Sci U S A. 2020; 117(45):28344-28354). Both mechanisms prevent the transcriptional induction of IFN-stimulated genes (ISGs) that establish an antiviral state by interfering with the nuclear import of STAT proteins. In addition, it was described that ORF6 from SARS-CoV-1 and SARS-CoV-2 inhibited nuclear export of host mRNAs, which downregulated protein expression of newly transcribed genes (Addetia et al. 2021. mBio. Vol. 12, 2; Kato et al. Biochem Biophys Res Commun. 2021;536:59-66).

Building on this previous knowledge, the manuscript analyzes with a new experimental approach, nucleus-cytoplasm cellular fractionation and mRNAseq, which cellular mRNA species are affected by inhibition of mRNA nuclear export when ORF6 was overexpressed. It was found that some antiviral factors, most of them ISGs, were enriched in the nucleus of cells expressing ORF6 vs cells expressing GFP protein as a control.

The manuscript provides further insight into the relevance of SARS-CoV-2 ORF6 in the interference with host antiviral responses. Characterizing the mechanisms of virus virulence will contribute to the identification of potential therapeutic targets. Therefore, the manuscript provides interesting novel information. Nevertheless, there are some concerns that should be addressed and are detailed below:

Specific points

1. Fig. 2A. In order to draw conclusions about Gag1 protein expression, the authors may consider to normalize Gag levels by the endogenous control HSP90.

2. Using mRNAseq, the paper shows that overexpression of ORF6 in human HEK293T cells led to 2,431 mRNA species enriched in the nucleus of ORF-6 vs GFP expressing cells. From these results, it is concluded that “ORF6 inhibits mRNA export of a broad range of cellular mRNAs”. However, the enrichment of these mRNAs in the nucleus might be caused alternatively by an increase in mRNA transcription in ORF6 vs GFP expressing cells. To exclude this possibility, cells were treated with IFN-ß, which, in principle, would promote the transcription of hundreds of IFN-stimulated genes (ISGs) and would allow to compare differences in mRNA export in ORF6 vs GFP-expressing cells. However, since ORF6 protein inhibits the transcriptional induction of ISGs, the interpretation of mRNAseq results in IFN-treated cells is confusing. To overcome this limitation, the authors might consider to use other exogenous factors that promote transcription in a similar way in ORF6 and GFP-expressing cells.

3. Lines 282-289. In a context of virus infection in Vero cells, several ISGs were analyzed by RT-qPCR both in the nuclear and cytoplasmic fractions. An enrichment of RIG-I and IRF1 (around 4-and 2-fold, respectively) was observed in the nucleus of SARS-CoV-2 infected vs non-infected cells, while other ISGs were not significantly enriched. Since Vero cells are deficient in IFN production, this cell line might not be the most appropriate to evaluate the inhibition of ISGs during infection and other cell lines such as Calu3 might be considered.

4. Line 290. From the results described in the previous point, it is concluded that “ORF6 can block the export of these mRNA species in the context of SARS-CoV-2 infection”. In addition to ORF6, SARS-CoV-2 is expressing during infection other genes with a known effect in the inhibition of mRNA nuclear export, such as nsp1 (Huang et al. J Virol. 2007). Therefore, ORF6 should not be considered the main determinant of mRNA export inhibition. To unequivocally determine whether ORF6 is responsible for this effect, a SARS-CoV-2-∆ORF6 mutant should be used.

Minor points

1. Lines 59-60. Introduction. The authors may want to revise the statement with the list of SARS-CoV-2 accessory proteins according to a recent work by Jungreis et al. (SARS-CoV-2 gene content and COVID-19 mutation impact by comparing 44 Sarbecovirus genomes. Nat Commun 12, 2642. 2021). This paper found that only ORFs 3a, 3c, 6, 7a, 7b, 8 and 9b show strong protein-coding evolutionary signature, whereas ORFs 3b, 9c, and 10 lack protein-coding signatures or convincing experimental evidence of protein-coding function.

2. Lines 66-67. Introduction. The authors may want to revise the statement that ORF6 is the most divergent accessory gene between SARS-CoV-1 and SARS-CoV-2 ORF6. ORF6 is highly conserved in both SARS-CoV-1 and -2, while ORF 8 is one of the most divergent accessory genes.

3. Line 69. Introduction. Please note that Reference #10 describes SARS-CoV-2 Orf6 interaction with cellular Nup98, instead of SARS-CoV-1 Orf6.

4. Line 235. mRNAs species that were significantly upregulated after IFN-β treatment were labelled “interferon upregulated genes (IUGs)”. To avoid misinterpretations, could these mRNAs be named IFN-stimulated genes (ISGs) as done by the majority of the scientific community?

**Part II – Major Issues: Key Experiments Required for Acceptance**

Reviewer #1: Major:

1. Throughout the manuscript ORF9b is used as a negative control however there is little expression of 9b in any of the western blots. This should be explained as you would want an expression control that expresses in these cells. For example, in Figure 1B, 2A, Supplementary data S1A. This should be commented on or fixed in the figures.

2. In Figure 1F where ORF6 mutants are being expressed, there are seemingly size differences on the western blot of the mutants whilst they should have minimal changes due to the mutants being made. Is there an explanation for this difference? Are they post translationally modified that explains the weight change?

3. In Figure 2, the western blots are much over exposed especially in Figure 2A, for example the Rae1 slice. Please provide less exposed images. Zooming in to the image shows just large black pixels leading to the idea that this is not an authentic image. The whole blot should be provided.

4. In Figure 2, and throughout the whole paper, there is no reference to the fact that ORF6 is membrane associated. Where immunoprecipitations are used, does the membrane tropism of ORF6 alter the interpretation of the data? Are appropriate conditions being used to IP out of membrane fractions? Based on the model at the end of the paper, there is no realization that ORF6 is membrane associated. It is not free-floating in the cytoplasm, to the contrary as shown in the Miorin et al paper, it is ER/Golgi associated. This should be discussed, and the model changed in the paper discussion.

5. Toward the idea of binding and localization, the Supplemental Figure 6 does show a co-localization experiment with Nup98 and ORF6. This should be quantified for association but based on this data there is little colocalization. Additional quantitation of co-localization with confocal microscopy and quantitation of overlap should be performed.

6. The experiments on the effects of exogenous RNAs use GFP as the reporter, as in the LMB treatment experiments in Supp Figure 3. There is discontinuity in the data showing the control LMB treatment of ORF6 co-transfected cells in S3E and the western blot in S3F. In S3E, there is little GFP expression in the ORF6 expressing cells while in the western blot in S3F, there is substantial GFP bands. This discrepancy should be discussed, or more representative data shown. Is the fluorescence and western blot from the same experiment?

7. In the same experiment, the quantitation of mRNA in the nucleus and cytoplasmic fraction is quantified. The ORF6 containing cells have reduced mRNA in the cytoplasm (Supp Figure 3G) but only 3-4 fold lower than the ORF9b plasmid transfected cells. Is this enough of a difference to explain the vastly different mRNA levels described in the text and quantified in Figure S3H?

8. In the RNASeq and fractionation section of the grant there are some analysis questions that should be further explained. The heatmap in Figure 4 has expression from +1 to -1. This is a small change for any regular RNASeq dataset analysis. Is this expected here? More discussion in the text is needed on the fold changes seen. Shown in the volcano plots in Figure 4, almost all of the genes of interest have a Log2 fold change of less than 1. This minor change should be explained in the text.

9. From this small difference in nuclear vs cytoplasmic difference, the fold changes seen in Figure 5 show some larger changes. The read counts of these genes should be provided in a table to show the read numbers between nucleus and cytoplasm. Are there large increases in these read numbers in total?

10. The localization data in Figure 5 is intriguing where Delta (B.1.617.2) has a dramatically different ORF6 localization compared to the other viruses however all of the viruses tested have identical ORF6 sequences. Do you have an explanation for this difference? As stated in line 281, one idea is the viability of the cells. If taken earlier in the infection, does the staining look like the other variants tested?

Reviewer #2: This is an elegant and technically sound study addressing a timely and important question under hot debate. Their findings are interesting and bringing in IRF1 and RIG-I, but there is a major issue to be addressed.

Major concerns:

Recent studies on SARS-CoV-2 ORF6 -NUP98-RAE1 complex interaction discovered by several groups are under hot debate.

Miorin et al. proposed that ORF6 binding to the NUP98-RAE1 complex inhibits the docking of KPNB1 to the NPC. Conversely, Kato et al. suggested that aberrant nucleocytoplasmic transport is caused by the ORF6 dislocating NUP98 from the NPCs to the cytoplasm.

For instance, a research group from Stony Brook Univ. suggested decoupling SARS-CoV-2 ORF6 localization and interferon antagonism (JCS on-line). Moreover, Timothy Mitchison also confirmed the dislocation of nuceloporins which were consistent with Kato et al., which was unlikely to be directly involved in the nuclear transport inhibition by Miorin et al. (bioRxiv preprint).

What is their opinion on this key issue? They should discuss the recent JCS and Mitchison ‘s bioRxiv and deliberating their opinions on Miorin et al. & Kato et al. findings with their data.

Reviewer #3: Since ORF6 protein inhibits the transcriptional induction of ISGs, the interpretation of mRNAseq results in IFN-treated cells is confusing. To overcome this limitation, the authors might consider to use other exogenous factors that promote transcription in a similar way in ORF6 and GFP-expressing cells.

In a context of virus infection in Vero cells, several ISGs were analyzed by RT-qPCR both in the nuclear and cytoplasmic fractions. An enrichment of RIG-I and IRF1 (around 4-and 2-fold, respectively) was observed in the nucleus of SARS-CoV-2 infected vs non-infected cells, while other ISGs were not significantly enriched. Since Vero cells are deficient in IFN production, this cell line might not be the most appropriate to evaluate the inhibition of ISGs during infection and other cell lines such as Calu3 might be considered.

**Part III – Minor Issues: Editorial and Data Presentation Modifications**

Reviewer #1: 1. Line 68: What does “This” refer to? It seemingly references SARS-CoV-1 whilst the statement on Rae1-Nup98 goes on about SARS-CoV-2’s ORF6.

2. Figure 1D: Add ORF9b to the list of genes that don’t express well.

3. Line 132: It is reference 9, not reference 8 to cite here.

4. Line 346: provide a reference for the statement that ORF6 is in the SARS-CoV-1 virion.

Reviewer #2: Not just mRNA export, Rae1-Nup98 also played key roles in cell cycle.

doi: 10.1038/nature04221. ;

doi: 10.1073/pnas.0807660105.;

doi: 10.1073/pnas.0609582104.

Reviewer #3: Indicated in the information to the authors

PLOS authors have the option to publish the peer review history of their article (what does this mean?). If published, this will include your full peer review and any attached files.

Reviewer #1: No

Reviewer #2: No

Reviewer #3: No
---

## [Decision Letter · Decision Letter 1]

21 Jul 2022

Dear dr. Bishop,

We are pleased to inform you that your manuscript 'SARS-CoV-2 ORF6 disrupts innate immune signalling by inhibiting cellular mRNA export' has been provisionally accepted for publication in PLOS Pathogens.

Best regards,

Bart L. Haagmans

Guest Editor

PLOS Pathogens

Ron Fouchier

Section Editor

PLOS Pathogens

Kasturi Haldar

Editor-in-Chief

PLOS Pathogens

orcid.org/0000-0001-5065-158X

Michael Malim

Editor-in-Chief

PLOS Pathogens

orcid.org/0000-0002-7699-2064

I am sorry for the delay but the reviewers now have submitted their comments. Based on these reviews I am happy to accept the manuscript for publication in Plos Pathogens. All reviewers indicated that you addressed their concerns and the neccessary changes were made in the revised manuscript. I think you also addressed one remaing issue, indicated by one of the reviewers, which is related to the localization of ORF6. By adding some references related to the membrane association of ORF6 in the discussion in your revised manuscript, no further modications to the figure are needed.

Reviewer Comments (if any, and for reference):

Reviewer's Responses to Questions

**Part I - Summary**

Reviewer #1: The authors have answered all of my comments pertaining to the critical areas of the paper. The only point I would request is to modify the language in the model section of the discussion to reiterate the membrane associated nature of ORF6 rather than showing it free in the cytoplasm as shown in the model figure.

Reviewer #2: I thank the authors for the substantial improvements of the manuscript. They have successfully addressed all my concerns. The manuscript is well-designed. It is an original research paper that should be interested for all

researches in the field of virology and cell biology. The revised manuscript has improved and can now be

accepted for publication.

Reviewer #3: The authors have satisfactorily addressed all my concerns and made the necessary changes to the manuscript. This new version of the manuscript has been significantly improved and, in my opinion, is now acceptable for publication.

**Part II – Major Issues: Key Experiments Required for Acceptance**

Reviewer #1: (No Response)

Reviewer #2: (No Response)

Reviewer #3: NA

**Part III – Minor Issues: Editorial and Data Presentation Modifications**

Reviewer #1: (No Response)

Reviewer #2: (No Response)

Reviewer #3: NA

PLOS authors have the option to publish the peer review history of their article (what does this mean?). If published, this will include your full peer review and any attached files.

Reviewer #1: No

Reviewer #2: No

Reviewer #3: No

---

## [Editor Report · Acceptance letter]

19 Aug 2022

Dear Dr. Bishop,

We are delighted to inform you that your manuscript, "SARS-CoV-2 ORF6 disrupts innate immune signalling by inhibiting cellular mRNA export," has been formally accepted for publication in PLOS Pathogens.

Best regards,

Kasturi Haldar

Editor-in-Chief

PLOS Pathogens

orcid.org/0000-0001-5065-158X

Michael Malim

Editor-in-Chief

PLOS Pathogens

orcid.org/0000-0002-7699-2064